# Optimal privacy guarantees for a relaxed threat model: Addressing sub-optimal adversaries in differentially private machine learning

**Georgios Kaissis** *
AI in Medicine and Healthcare
Technical University of Munich
Munich, Germany
`g.kaissis@tum.de`

**Alexander Ziller**
AI in Medicine and Healthcare
Technical University of Munich
Munich, Germany
`alex.ziller@tum.de`

**Stefan Kolek**
Mathematical Foundations of AI
LMU Munich
Munich, Germany
`kolek@math.lmu.de`

**Anneliese Riess**
Machine Learning in Biomedical Imaging
Helmholtz Munich
Neuherberg, Germany
`anneliese.riess@helmholtz-munich.de`

**Daniel Rueckert**
AI in Medicine and Healthcare
Technical University of Munich
Munich, Germany
`daniel.rueckert@tum.de`

## Abstract

Differentially private mechanisms restrict the membership inference capabilities of powerful (optimal) adversaries against machine learning models. Such adversaries are rarely encountered in practice. In this work, we examine a more realistic threat model relaxation, where (sub-optimal) adversaries lack access to the exact model training database, but may possess related or partial data. We then formally characterise and experimentally validate adversarial membership inference capabilities in this setting in terms of hypothesis testing errors. Our work helps users to interpret the privacy properties of sensitive data processing systems under realistic threat model relaxations and choose appropriate noise levels for their use-case.

## 1 Introduction

Machine learning (ML) offers a powerful set of techniques for addressing complex problems across various fields, including fields where sensitive data is processed, e.g. medicine or finance. However, the use of sensitive data in ML comes with privacy challenges: It is known that ML models memorise their training data [1], and that some degree of memorisation may be necessary for the best model performance [2]. When a model memorises private information, it can reveal this information when attacked by adversaries. A very important class of attack is *Membership Inference* (MI) [3], which aims to determine whether the data of a specific individual was used to train a machine learning model. MI can lead to the disclosure of highly sensitive information about the individual, e.g. that

---

*GK is also with the Institute of Machine Learning in Biomedical Imaging, Helmholtz Munich and the Department of Computing, Imperial College London.

37th Conference on Neural Information Processing Systems (NeurIPS 2023).

they are part of a database of cancer patients which was used to train a predictive medical ML system. The importance of studying MI is furthermore derived from the fact that a successful MI attack signifies that the model is also vulnerable to all other attacks such as data reconstruction or attribute inference ("most significant bit" property [4]).

Differential Privacy (DP) [5] is a formal framework and collection of techniques to furnish objective privacy guarantees for ML model training. It is a natural counterpart to MI, and it is possible to interpret DP entirely through the MI lens, which is known as the hypothesis testing interpretation of DP [6, 7]. Given a database $D$ and a database $D'$ which differ in exactly one record, hypothesis testing DP quantifies the trade-offs between the Type-I and Type-II errors of an adversary using a hypothesis test to determine whether a model was trained on $D$ or on $D'$. This hypothesis test is considered to be optimal in the sense of the Neyman-Pearson (NP) lemma [8], which makes the *DP Threat Model* (DPTM) very strong: It assumes that the adversary has full access to the training database, knowledge of the specifications of the DP mechanism and can even deeply manipulate the process of ML model training itself to their advantage [9]. Although this has the benefit that DP is a true "worst-case" guarantee and thus holds for all weaker adversaries, it may be unnecessarily conservative in practice. For instance, the assumption that the adversary has full access to the training data may be unrealistic in settings where this data is tightly access-restricted. As a consequence, previous works also considered threat models with weaker adversaries, i.e. *Relaxed Threat Models* (RTMs), and experimentally showed that the empirical protection against MI offered by DP mechanisms is much stronger than the upper bound predicted by the DP analysis [9, 10, 11]. While such empirical investigations provide valuable initial insights, we contend that an additional *formal* analysis of specific, practically relevant RTMs can greatly benefit stakeholders in conducting a thorough privacy analysis of ML systems. For example, a "privacy certificate" for an ML model can include both (1) the MI risk under the DPTM (which holds when "all bets are off") but also (2) the more realistic MI risk under whichever RTM is applicable to the actual model training conditions.

In this work, we study MI in the context of a specific RTM with high practical relevance, in which the adversary has *incomplete access to the model's training data*. An example of this RTM is the training of a predictive ML model on medical data which is generated and stored at a single hospital and never released, whereas the model is shared with third parties. Here, an MI adversary must resort to leveraging auxiliary data from the same distribution as the hospital's target database. A similar attack (*offline MI*) was studied in [12, 13, 14], and also does not use the full database for the attack (albeit for reasons of computational efficiency). As MI is equivalent to a *hypothesis testing* problem, the aforementioned works demonstrate that the Type-I and Type-II errors (i.e. the false positive/false negative rates) of offline MI attacks are higher than MI attacks assuming the DPTM. In this work, we formally derive and validate exact bounds on the error rates of MI attacks against DP noise mechanisms in the aforementioned RTM (which we will refer to as "the RTM" from now on).

**Contributions**   Our contributions are as follows: (1) Since MI can be fully characterised using hypothesis testing, we begin by formalising the RTM as a hypothesis testing security game and show that the RTM adversary is strictly sub-optimal in terms of error rates compared to the DPTM adversary; (2) Next, we formally derive the "best sub-optimal" hypothesis testing MI strategy in the RTM; (3) We then analyse the error rates associated with this strategy against common additive noise mechanisms used in DP under the RTM. (4) We express these error rates using trade-off functions, which parameterise all achievable Type-I/Type-II error combinations, similar to the approach taken by [7] in defining f-DP. (5) Additionally, we introduce a new metric, $\lambda$-Detection Resilience ($\lambda$-DR), which allows us to compare the hypothesis testing capabilities of the RTM and DPTM adversaries, despite their differing threat models. (6) We conclude our study by empirically auditing the proposed bounds and demonstrating accuracy advantages in the training of deep neural networks with DP-SGD.

## 2   Background

**Related work**   Our work contributes to the ongoing research in three distinct, yet related areas of privacy-preserving ML. Firstly, we aim to augment empirical investigations which use MI to establish lower bounds on the privacy guarantees of ML systems, with formal guarantees. For instance, [9] experimentally evaluate several types of adversaries against DP ML workflows, [10, 11] present empirical privacy analyses of DP-SGD [15], while [16] focus on federated learning. Such empirical privacy auditing techniques have found their way into software tools such as *ML-Doctor* [17] or *ML*

*Privacy Meter* [14]. As mentioned above, our RTM considers an adversary performing MI without access to the (complete) model's training data. This type of MI attack has been studied in several prior works, most notably by Watson et al. [13], Ye et al. [14] and Carlini et al. [12], who find the adversary's hypothesis testing capabilities to be substantially diminished compared to the DPTM. The second line of work related to our paper are formal analyses of MI in the DPTM, such as [18], who consider a Bayesian adversary or [4], who bypass the DP analysis and directly study the properties of the underlying distributions. Moreover, [19] and [20] are rooted in hypothesis testing theory like our work, but only consider threat models equivalent to the DPTM, while [21] consider restricted adversaries, albeit not through the hypothesis testing lens. Lastly, we adopt the recent development of using functions (rather than numbers) to parameterise privacy guarantees, which is becoming common practice in the DPTM [7, 22, 23], but has not yet been used to study threat model relaxations. Our paper unifies these strands of work by introducing a formal guarantee in the RTM, which allows for a more nuanced analysis of the privacy risks associated with machine learning models.

**Differential privacy**   DP [5] is a stability condition on randomised mechanisms requiring that their outcomes should be approximately equally likely when the data of a single individual is added or removed from the input database. Throughout this work, we will consider the scenario of ML model training, where DP is realised through an *additive noise mechanism*. We consider a deterministic model training function $q$. The global sensitivity $\Delta > 0$ of $q$ is defined as $\Delta = \sup_{D,D':D\simeq D'} \|q(D) - q(D')\|$ where the $\sup$ is taken over all pairs of neighbouring databases $D \simeq D'$ which differ in a single record. [2] $\Delta$ is measured in some norm, which we will indicate by subscript wherever relevant (e.g. $\Delta_1$). An additive noise mechanism (or just mechanism) $\mathcal{M}$ maps databases and probability distributions to a (model) parameter space $\Theta$. Its outputs are of the form $\theta = q(X) + z, z \sim \mathcal{Z}$, where $\mathcal{Z}$ is a noise distribution. Since $q$ and the database are considered deterministic, $\theta$ is a random variable induced by the randomness of $\mathcal{Z}$. Unless otherwise indicated, $\mathcal{M}$ will be realised through distributions $\mathcal{Z}(\mu, \xi)$ supported on the real line which admit a density. We assume that $\xi$ is fixed and public information and that $\mathcal{Z}$ is shift invariant. In other words, $\mathcal{M}(D) = (q(D) + \mathcal{Z}(0,\xi)) \sim \mathcal{Z}(q(D),\xi)$ and $\mathcal{M}(D') = (q(D') + \mathcal{Z}(0,\xi)) \sim \mathcal{Z}(q(D'),\xi)$.

In the hypothesis testing interpretation of DP [6, 7, 24], the adversary attempts to determine whether $\theta$ was trained on $D$ or $D'$ using a classical/frequentist hypothesis test. The adversary considers the hypotheses $\mathcal{H}_0 : \theta \sim \mathcal{M}(D)$ vs. $\mathcal{H}_1 : \theta \sim \mathcal{M}(D')$ and their reverse, i.e. $\mathcal{H}_0 : \theta \sim \mathcal{M}(D')$ vs. $\mathcal{H}_1 : \theta \sim \mathcal{M}(D)$. $\mathcal{H}_0$ and $\mathcal{H}_1$ are called the *null* and *alternative* hypothesis, respectively. The aim of the hypothesis testing problem is to distinguish between $\mathcal{M}(D)$ and $\mathcal{M}(D')$ using a pair of decision rules (tests). Let $\phi, \phi'$ be tests with Type-I error rate $\alpha_\phi = \mathbb{E}_{\mathcal{M}(D)}(\phi)$ and Type-II error rate $\beta_\phi = 1 - \mathbb{E}_{\mathcal{M}(D')}(\phi)$ (identical for $\phi'$). The adversary's goal will be to maximise their *power* $1 - \beta$ at a fixed level $\alpha$, since the fundamental trade-off in hypothesis testing is that $\alpha$ and $\beta$ cannot be minimised simultaneously [25]. All achievable $\alpha/\beta$ pairs for a given adversary/test combination are expressed by a pair of functions on the unit square, called the *trade-off functions* $T$ and $T^{-1}$.

$T$ is defined through the test $\phi$: $\mathcal{M}(D)$ vs. $\mathcal{M}(D')$ as $T(\mathcal{M}(D), \mathcal{M}(D'))(\alpha) = \inf_\phi \{\beta_\phi : \alpha_\phi \leq \alpha\}$, whereas $T^{-1} = T(\mathcal{M}(D'), \mathcal{M}(D))$ is defined using $\phi'$. The $\inf$ is taken over the set of tests which can be constructed in a given threat model, in this case the DPTM. For mechanisms with symmetric noise density, $T = T^{-1}$, allowing us to consider only a single trade-off function. In general, one takes the symmetrisation/convexification of $T$ and $T^{-1}$, denoted $\mathrm{C}(T, T^{-1})$ which creates a single, symmetric trade-off function. Since trade-off functions fully characterise the adversary's error rates, they can be used to compare the privacy guarantees of different mechanisms and/or different threat models. In particular, the closer the trade-off function's graph is to the line $\beta(\alpha) = 1 - \alpha$, i.e. the off-diagonal of the unit square, the stronger the implied privacy guarantee of the mechanism is. The notion of expressing DP guarantees based on a comparison of $\mathrm{C}(T, T^{-1})$ with a "reference" trade-off function $f$ is called f-DP [7]:

**Definition 1.** *A mechanism $\mathcal{M}$ satisfies f-DP, if $\forall D, D' : D \simeq D'$ it holds that $\mathrm{C}(T, T^{-1})(\alpha) \geq f(\alpha) \forall \alpha \in [0, 1]$ for a reference trade-off function $f$ (see Figure 2 of [7] for an example).*

*A pair of distributions $\mathcal{Z}(\mu_1, \xi), \mathcal{Z}(\mu_2, \xi)$ is called a dominating pair [22] for $\mathcal{M}$ if, $\forall D, D' : D \simeq D'$, $\mathrm{C}(T, T^{-1})(\alpha) \geq \mathrm{C}(T(\mathcal{Z}(\mu_1, \xi), \mathcal{Z}(\mu_2, \xi)), T(\mathcal{Z}(\mu_2, \xi), \mathcal{Z}(\mu_1, \xi)))(\alpha) \forall \alpha \in [0, 1]$.*

---

[2] In particular, we will assume databases are true sets and that $D'$ is constructed from $D$ by adding or removing a single record. The neighbouring relation is denoted $\simeq$.

For example, the dominating pair for the Gaussian Mechanism (GM) is $(\mathcal{N}(0, \sigma^2), \mathcal{N}(\Delta_2, \sigma^2))$ and for the Laplace Mechanism (LM), it is $(\text{Lap}(0, b), \text{Lap}(\Delta_1, b))$. The dominating pair distributions thus exhibit the greatest *effect size* (in this case $\Delta_1/b$ or $\Delta_2/\sigma$), allowing for the construction of the trade-off functions whose graphs are farthest from the off-diagonal of the unit square. In other words: the construction of the trade-off functions is determined by the hypothesis tests with the greatest power $1 - \beta$ under a specific threat model, evaluated at the dominating pair distributions.

In the DPTM, $\mathcal{H}_0$ and $\mathcal{H}_1$ are both *simple hypotheses* of the form $\mathcal{H}_0 : \theta \in \Theta_0 \subset \Theta$ vs. $\mathcal{H}_1 : \theta \in \Theta_1 \subset \Theta$ with $\|\Theta_0\| = \|\Theta_1\| = 1$ and $\Theta_0 \cap \Theta_1 = \emptyset$. The reason is that the distributions of $\mathcal{M}(D)$ and $\mathcal{M}(D')$ are fully specified because the adversary can evaluate $q(D)$ and $q(D')$. The NP lemma [8] states that, for simple vs. simple hypothesis tests such as the ones in the DPTM, the *uniformly most powerful* (UMP) level $\alpha$ tests $\phi$ and $\phi'$ (with the highest power $1 - \beta$ at a specified level $\alpha$ over the entire range of $\theta$, in this case $q(D)$ or $q(D')$) can be constructed by thresholding the (log-) Likelihood Ratio (LRs) $\Lambda/\Lambda'$ of the mechanism's outputs $\theta$ under $D$ and $D'$:

$$\Lambda(\theta) = \log \frac{\mathcal{L}(\theta \mid \mathcal{M}(D'))}{\mathcal{L}(\theta \mid \mathcal{M}(D))} \lessgtr c \text{ and } \Lambda'(\theta) = \log \frac{\mathcal{L}(\theta \mid \mathcal{M}(D))}{\mathcal{L}(\theta \mid \mathcal{M}(D'))} \lessgtr c,$$

where $c \in \mathbb{R}$ is the critical value. The trade-off functions in the DPTM are then constructed from the distributions of $\Lambda/\Lambda'$ evaluated at the dominating pairs under the respective $\mathcal{H}_0$ and $\mathcal{H}_1$ (also called *privacy loss distributions* [23]). Since the DPTM adversary enjoys the optimality properties of the NP lemma, we will refer to this adversary as $\mathcal{A}^{\text{NP}}$.

## 3 Hypothesis testing in the RTM

The trade-off function's form for a given threat model depends only on the test statistic's distributions under $\mathcal{H}_0$ and $\mathcal{H}_1$. Our strategy for the rest of the paper is thus: (1) Derive the optimal tests the adversary can construct in the RTM; (2) Specify the test statistic distributions for these tests, and (3) Construct the trade-off functions for a given mechanism using these test statistics, which will allow us to tightly bound the adversary's error rates and compare them to the DPTM. We first recall a general result, specifying how trade-off functions are constructed from test statistics.

**Definition 2** (Trade-off function construction). *Let $P, Q$ be the distributions of a test statistic under $\mathcal{H}_0$ and $\mathcal{H}_1$, respectively and let $\Phi_{P/Q}$ denote their cumulative distribution function (CDF), $\Psi_{P/Q}$ their survival function (SF) and $\Phi_{P/Q}^{-1}, \Psi_{P/Q}^{-1}$ their respective inverses (iCDF, iSF). Then, we define:*

$$T(\alpha) = \Phi_Q(\Psi_P^{-1}(\alpha)) \text{ and } T^{-1}(\alpha) = \Psi_P(\Phi_Q^{-1}(\alpha)). \tag{1}$$

For example, in experimental studies, empirical proxies of the test statistic distributions are evaluated by training *shadow models* [3]. Then, the empirical trade-off functions are constructed, which parameterise the error rates of the empirical adversary under a specific threat model [11, 12, 14]. We will utilise this strategy later to validate our theoretical bounds empirically. For our formal analysis on the other hand, we will derive exact expressions for $T$ and $T^{-1}$.

**RTM security game** For reasons which will soon become clear, we will refer to the adversary under our RTM as a *sub-optimal adversary* ($\mathcal{A}^{\text{SO}}$) and use a formal security game, similar to the one used to define the DPTM (see e.g. Section 3.1. of [4]), which proceeds between a neutral/trusted curator $\mathcal{C}$ and $\mathcal{A}^{\text{SO}}$. We consider a single-round, non-interactive protocol.

Step 1: The adversary $\mathcal{A}^{\text{SO}}$ constructs a database $D = \{x_1, \dots x_n\} \subseteq \mathbb{D}$, decides on a training function $q$, a DP mechanism $\mathcal{M}$ and sends them to the curator $\mathcal{C}$;

Step 2: $\mathcal{C}$ flips a bit $b$. If $b = 0$, they fix $X = D$. If $b = 1$, they choose a singleton $\{x^*\} \subset \mathbb{D}'$, where $\mathbb{D} \subset \mathbb{D}'$ and $\mathbb{D} \cap \mathbb{D}' = \{x^*\}$ and fix $X = D \cup \{x^*\}$. They then train a model $\theta \in \Theta$ on $X$ using $q$ and $\mathcal{M}$ ($\theta$ may also be a gradient) and send $\theta$ to $\mathcal{A}^{\text{SO}}$;

Step 3: $\mathcal{A}^{\text{SO}}$ decides if $\mathcal{H}_0 : \theta \sim \mathcal{M}(D)$ or $\mathcal{H}_1 : \theta \sim \mathcal{M}(D')$;

Step 4: If $\mathcal{A}^{\text{SO}}$ is correct, the MI attack is successful, privacy is breached and the game is won.

The key difference between the RTM and the DPTM is step 2: $\mathcal{A}^{\text{SO}}$ has *no access to the point $x^*$* which may be added by $\mathcal{C}$, and by extension, no knowledge of the exact value of $q(D')$. In step 3, $\mathcal{A}^{\text{SO}}$ can therefore only decide $\theta \sim \mathcal{M}(D')$ by rejecting that $\theta \sim \mathcal{M}(D)$, but not by directly confirming that $\theta \sim \mathcal{M}(D')$.

The hypothesis testing capabilities of the RTM adversary are identical to those of an *offline MI adversary*. Offline MI attacks are used in previous works to audit the privacy of ML model training [12, 13, 14] and decide whether $x^*$ is part of the training data or not without actively training shadow models on databases containing it. Equivalently, offline MI is *MI by exclusion* (i.e. inferring $\theta \sim \mathcal{M}(D')$ by rejecting $\mathcal{H}_0 : \theta \sim \mathcal{M}(D)$). In offline MI, this is done for reasons of practicality and computational efficiency [12], while in the RTM, it by design. Nonetheless, the resulting error rates are identical and our bounds thus can be used to characterise offline MI.

**RTM: Formal analysis**  $\mathcal{M}$ is an additive noise mechanism based on the distribution $\mathcal{Z}$, thus $\mathcal{H}_0 : \theta \sim \mathcal{M}(D)$ is equivalent to $\mathcal{H}_0 : \theta \sim \mathcal{Z}(q(D), \xi)$. Since $q(D)$ is computable by $\mathcal{A}^{\text{SO}}$ through access to $D$, the likelihood $\mathcal{L}(\theta \mid \mathcal{Z}(q(D), \xi))$ is also computable and $\mathcal{H}_0$ is a simple hypothesis of the form $\theta \in \Theta_0, \|\Theta_0\| = 1$, exactly like the DPTM. However, the likelihood $\mathcal{L}(\theta \mid \mathcal{Z}(q(D'), \xi))$ is *not computable* without access to $D'$ and $\mathcal{H}_0$ is a *composite hypothesis* of the form $\theta \in \Theta \setminus \Theta_0$ with $\|\Theta \setminus \Theta_0\| > 1$ because it depends on the unknown value of $q(D')$. The following is a direct consequence of the NP lemma, which holds only for simple vs. simple hypotheses [25].

**Lemma 1.** *In the RTM, no UMP level $\alpha$ test exists for all possible values of $q(D')$, i.e. $\forall \theta \in \Theta \setminus \Theta_0$. Thus, $\mathcal{H}_0$ vs. $\mathcal{H}_1$ (and $\mathcal{H}_1$ vs. $\mathcal{H}_0$) are each not decidable by a single test.*

This implies that $\mathcal{A}^{\text{SO}}$ is necessarily *weaker* than $\mathcal{A}^{\text{NP}}$, i.e. cannot achieve the same power $1 - \beta$ at a given level $\alpha$ for all values of the parameter of interest; to thus establish the *optimal tests in this sub-optimal setting*, $\mathcal{A}^{\text{SO}}$ must consider specific value ranges of $q(D')$ separately. In the following, we will call two tests *equivalent* if they have the same power at a level $\alpha$ for all values of the tested parameter. Moreover, we will limit our description to $\mathcal{H}_0 : \theta \sim \mathcal{M}(D)$ vs. $\mathcal{H}_1 : \theta \sim \mathcal{M}(D')$, as $\mathcal{H}_1$ vs. $\mathcal{H}_0$ is handled identically.

**Lemma 2.** *The composite LR test $\mathcal{H}_0 : \theta \sim \mathcal{Z}(q(D), \xi)$ vs. $\mathcal{H}_1 : \theta \sim \mathcal{Z}(q(D'), \xi)$ is equivalent to letting $\theta \sim \mathcal{Z}(\mu, \xi)$ and simultaneously conducting two individual one-sided LR tests $r$ and $r'$ with null hypothesis $\mu = 0$ and alternative hypotheses $-\Delta \le \mu < 0$ for $r$ and $0 < \mu \le \Delta$ for $r'$, where $\mu = q(D') - q(D)$ and $\Delta$ is the global sensitivity of $q$.*

Thus, $\mathcal{A}^{\text{SO}}$ can leverage known facts about $q$ and $\mathcal{M}$ to limit the value ranges for the alternative hypothesis, but, as $q(D')$ is unknown, must "split the problem" into two individual tests. The individual tests are then UMP level $\alpha$ under the following pre-condition.

**Lemma 3.** *The individual LR tests $r$ and $r'$ are UMP level $\alpha$ if the noise distribution $\mathcal{Z}$ has the monotone likelihood ratio property (MLRP). The Laplace Mechanism (LM), Gaussian Mechanism (GM) and the Poisson-Subsampled Gaussian Mechanism (SGM) all have the MLPR. Moreover, the power of both tests is maximised when $\|\mu\| = \|q(D') - q(D)\| = \Delta$.*

This lemma intuitively states that, while there is an optimality condition associated with the tests that can be constructed in the RTM, it is not the same optimality condition as the NP lemma guarantees in the DPTM. In particular, although $\mathcal{A}^{\text{SO}}$ can construct a pair of tests which are optimal (UMP level $\alpha$), they are *only optimal for specific parameter value ranges* rather than for all possible parameter values. The lemma also states that it suffices to consider the tests $\mu = 0$ vs. $\mu = \pm \Delta$ when constructing the trade-off functions, as these correspond to the *highest effect size* at a fixed $\xi$. In other words, *dominating pairs remain dominating pairs* in the RTM. Next, we provide two convenience lemmata, which allow $\mathcal{A}^{\text{SO}}$ to conduct both tests simultaneously.

**Lemma 4.** *If the density of $\mathcal{Z}$ is additionally symmetric about the location parameter, the simultaneous individual LR tests $r$ and $r'$ are equivalent to performing a single test using the magnitude of the observation as a test statistic, i.e. letting $\|\theta\| \sim \|\mathcal{Z}(\mu, \xi)\|$ and testing $\mathcal{H}_0 : \|\mu\| = 0$ vs. $\mathcal{H}_1 : \|\mu\| > 0$. The power of this "combined" test is maximised at $\|\mu\| = \Delta$.*

**Lemma 5.** *For a (not necessarily symmetric) $\mathcal{Z}$ the Generalised Likelihood Ratio Test (GLRT) using the test statistic $\log \frac{\mathcal{L}(\theta|\mathcal{Z}(\hat{\mu}, \xi))}{\mathcal{L}(\theta|\mathcal{Z}(0, \xi))}$, where $\hat{\mu}$ is the maximum likelihood estimate of $\mu$ is equivalent to simultaneously conducting $r$ and $r'$. The GLRT's power is maximised at $\hat{\mu} = \Delta$.*

We stress that, while the two tests are UMP level $\alpha$ and can be "combined" into one test, the combined test is *not UMP level $\alpha$* over the whole range (by Lemma 1). To construct both trade-off functions, the steps above are either repeated for $\mathcal{H}_1$ vs. $\mathcal{H}_0$, or the trade-off function for $\mathcal{H}_0$ vs. $\mathcal{H}_1$ is inverted directly, i.e. in closed form or numerically.

$\lambda$**-Detection resilience**  Next, we introduce a measure for $\mathcal{A}^{\mathrm{SO}}$'s optimal achievable error rates under the RTM, inspired by f-DP: For each mechanism, we create a pair of trade-off functions $j$ and $j^{-1}$. These parameterise the error rates of the UMP level $\alpha$ tests against the dominating pair distributions of a given mechanism $\mathcal{M}$ in the RTM. Next, we "unify" the functions by symmetrisation/convexification to obtain a single, symmetric trade-off function $J : \alpha \mapsto \beta$, which expresses *all* optimal error rates of $\mathcal{A}^{\mathrm{SO}}$ against $\mathcal{M}$. Finally, we compare $J$ to a reference trade-off function $\lambda$. If $J$ lies on or above $\lambda$, $\mathcal{A}^{\mathrm{SO}}$ cannot achieve lower $\beta$ for any $\alpha$ in the task of detecting the presence of $x^*$ in $D'$ by observing $\theta$, and we say that $\mathcal{M}$ satisfies $\lambda$-Detection Resilience ($\lambda$-DR).

**Definition 3.** *Let $r \in \mathscr{R}$ be a test for $\mathcal{M}(D)$ vs. $\mathcal{M}(D')$ and $r' \in \mathscr{R}$ be a test for $\mathcal{M}(D')$ vs. $\mathcal{M}(D)$, where $\mathscr{R}$ is the set of LR tests which are computable in the RTM. To construct J, fix a level $\alpha$, then obtain $j = \inf_{r \in \mathscr{R}}\{\beta_r : \alpha_r \leq \alpha\}$ and $j^{-1} = \inf_{r' \in \mathscr{R}}\{\beta_{r'} : \alpha_{r'} \leq \alpha\}$. Then compute the symmetrisation/convexification $J = \mathrm{C}(j, j^{-1})$. We say that a mechanism $\mathcal{M}$ satisfies $\lambda$-DR, if for some reference function $\lambda$ and $\forall D, D' : D \simeq D'$, it holds that $J(\alpha) \geq \lambda(\alpha) \, \forall \alpha \in [0,1]$.*

We remark that computing the symmetrisation/convexification $\mathrm{C}(j, j^{-1})$ is only a convenience measure to circumvent the necessity to work with both $j$ and $j^{-1}$ separately. The concrete algorithm for computing C for any trade-off function pair can be found in Definition E.1 of the Appendix to [7].

Since $\mathcal{A}^{\mathrm{SO}}$ is weaker than $\mathcal{A}^{\mathrm{DP}}$, we expect the trade-off functions characterising their error rates to lie on or above the trade-off functions for the same mechanism under the DPTM (see Figure 1). However, it is important to remark that $\lambda$-DR is *not a DP guarantee*, although the opposite holds true.

**Lemma 6.** *If a mechanism satisfies $f$-DP, it also satisfies $f'$-DR, with $f'(\alpha) \geq f(\alpha) \, \forall \alpha \in [0,1]$.*

Moreover, since $\lambda$-DR is defined conditional on a restriction of $\mathcal{A}^{\mathrm{SO}}$'s background knowledge, it is not resilient to post-processing through auxiliary information. However, a weaker condition holds:

**Lemma 7.** *If a mechanism satisfies $f$-DP and $f'$-DR with $f' \geq f$, arbitrary post-processing can only deteriorate privacy up to the $f$-DP guarantee.*

It follows that $\lambda$-DR is *not closed under adaptive composition*. In fact, no privacy relaxation assuming restricted background knowledge is closed under adaptive composition [26]. However, mechanism-specific guarantees can be given for *non-adaptive composition*, where the data/mechanism parameters are fixed in advance and all intermediate models/gradients are released to $\mathcal{A}^{\mathrm{SO}}$. This covers the standard DP-SGD setting and is studied below.

## 4  Mechanism-specific analysis

Since precisely quantifying DR for a specific noise mechanism requires knowledge of the RTM trade-off functions, we now provide methods to compute $j$ and $j^{-1}$ for various DP mechanisms. The construction of $J = \mathrm{C}(j, j^{-1})$ is independent of the specific form of $j, j^{-1}$. By definition for all $j, j^{-1}$: $j(0) = j^{-1}(0) = 1, j(1) = j^{-1}(1) = 0$ and $\alpha \in [0,1]$.

**Laplace Mechanism**  The LM with scale $b$ on a function $q$ with global $\ell_1$-sensitivity $\Delta_1$ outputs $q(X) + \mathrm{Lap}(0, b)$. The LM has a symmetric noise density, thus, the magnitude distributions can be used as test statistics according to Lemma 4. Under $\mathcal{H}_0$, the test statistic follows an exponential distribution, while under $\mathcal{H}_1$, it follows the (uncommon) *folded Laplace* distribution [27]. Both distributions admit closed-form expressions for their CDF, SF, iCDF and iSF, thus the trade-off functions can also be characterised in closed form.

**Theorem 1.** *Let $\mu_1 = \Delta_1/b$. The LM satisfies $\mathrm{C}(j_{\mathrm{Lap}}(\alpha), j_{\mathrm{Lap}}^{-1}(\alpha))$-DR with:*

$$j_{\mathrm{Lap}}(\alpha) = \begin{cases} -\exp\left(-\mu_1\right)\sinh\left(\log\left(\alpha\right)\right), & \alpha \geq \exp\left(-\mu_1\right) \\ 1 - \alpha\cosh\left(\mu_1\right), & \textit{otherwise,} \end{cases} \tag{2}$$

*and*

$$j_{\mathrm{Lap}}^{-1}(\alpha) = \begin{cases} \frac{1}{\alpha\exp\left(\mu_1\right) + \sqrt{\alpha^2\exp\left(2\mu_1\right)+1}}, & \alpha < 1/2 - 1/2\exp\left(-\mu_1\right) \\ -\frac{\alpha-1}{\cosh\left(\mu_1\right)}, & \textit{otherwise.} \end{cases} \tag{3}$$

**Gaussian Mechanism** The GM with covariance matrix $\sigma^2 \mathbf{I}^d$ on a $d$-dimensional function $q$ with global $\ell_2$ sensitivity $\Delta_2$ outputs $q(\boldsymbol{X}) + \mathcal{N}(0, \sigma^2 \mathbf{I}^d)$. Like the LM, it also has a symmetric noise density, thus the (squared) magnitude distributions can be used as a test statistic. Let $\chi_d^2(0, \sigma^2)$ be the chi-squared distribution with $d$ degrees of freedom, i.e. the distribution of $\|\mathcal{N}(0, \sigma^2 \mathbf{I}^d)\|_2^2$. Let $\chi_d^2\left(\mu_2^2, \sigma^2\right)$ be the noncentral chi-squared distribution with $d$ degrees of freedom and noncentrality parameter $\mu_2^2 = \Delta_2^2/\sigma^2$, i.e. the distribution of $\|\mathcal{N}(\boldsymbol{\nu}, \sigma^2 \mathbf{I}^d)\|_2^2$ with $\|\boldsymbol{\nu}\|_2 = \Delta$. As in Definition 2, $\Phi, \Psi, \Phi^{-1}$ and $\Psi^{-1}$ denote the CDF, SF, iCDF and iSF of the subscripted distribution, respectively.

**Theorem 2.** *Let* $\mu_2 = \Delta_2/\sigma$. *The GM satisfies* $\mathrm{C}(j_{\mathrm{GM}}(\alpha), j_{\mathrm{GM}}^{-1}(\alpha))$*-DR with:*

$$j_{\mathrm{GM}}(\alpha) = \Phi_{\chi_d^2(\mu_2^2, \sigma^2)}\left(\Psi_{\chi_d^2(0,\sigma^2)}^{-1}(\alpha)\right) \quad and \quad j_{\mathrm{GM}}^{-1}(\alpha) = \Psi_{\chi_d^2(0,\sigma^2)}\left(\Phi_{\chi_d^2(\mu_2^2,\sigma^2)}^{-1}(\alpha)\right). \quad (4)$$

These functions have no general analytic representation, but are easy to evaluate numerically. However, an analytic representation is available for $j(\alpha)$ at $d = 1$. We discuss the importance of $d = 1$ below.

**Corollary 1.** *For* $d = 1$, $j(\alpha)$ *admits the following closed-form representation:*

$$j_{\mathrm{GM}}(\alpha \mid d = 1) = \Phi_{\mathcal{N}(0,1)}\left(\Psi_{\mathcal{N}(0,1)}^{-1}(\alpha/2) - \mu_2\right) - \Psi_{\mathcal{N}(0,1)}\left(\Psi_{\mathcal{N}(0,1)}^{-1}(\alpha/2) + \mu_2\right). \quad (5)$$

Next, we study non-adaptive composition, where a database and mechanism parameters are fixed ahead of time, and all intermediate results are released to $\mathcal{A}^{\mathrm{SO}}$. This setting is often encountered, e.g. in standard DP-SGD.

**Lemma 8.** *Let* $\mathrm{GM}_a, \mathrm{GM}_b$ *be GMs with noise variances* $\sigma_a^2 \mathbf{I}^d, \sigma_b^2 \mathbf{I}^d$ *on functions with sensitivities* $\Delta_{2a}, \Delta_{2b}$, *respectively. Then, the non-adaptively composed mechanism* GMC *has trade-off functions:*

$$j_{\mathrm{GMC}}(\alpha) = \Phi_{\chi_d^2(\kappa_c, \sigma_c^2)}\left(\Psi_{\chi_d^2(0,\sigma_c^2)}^{-1}(\alpha)\right) \quad and \quad j_{\mathrm{GMC}}^{-1}(\alpha) = \Psi_{\chi_d^2(0,\sigma_c^2)}\left(\Phi_{\chi_d^2(\kappa_c, \sigma_c^2)}^{-1}(\alpha)\right). \quad (6)$$

*with* $\kappa_c = (\Delta_{2a} + \Delta_{2b})^2/\sigma_a^2 + \sigma_b^2$ *and* $\sigma_c^2 = \sigma_a^2 + \sigma_b^2/4$.

The GM moreover exhibits specific asymptotic behaviour when the query dimensionality $d$ and/or the number of non-adaptive compositions increase.

**Theorem 3** (Blessing of dimensionality in the RTM). *Consider a GM on a function with sensitivity* $\Delta_2$ *and noise variance* $\sigma^2 \mathbf{I}^d$ *such that* $\Delta_2/\sigma \ll 1$. *Let* $\mu_2 = \Delta_2/\sigma$. *As* $d$ *and/or as the number of non-adaptive compositions* $N$ *increase,* $j_{\mathrm{GM}}$ *and* $j_{\mathrm{GM}}^{-1}$ *tend to the common form:*

$$\Phi_{\mathcal{N}(0,1)}\left(\frac{\Psi_{\mathcal{N}(0,1)}^{-1}(\alpha)}{\sqrt{\frac{2N\mu_2^2}{d} + 1}} - \frac{\sqrt{2}N\mu_2^2}{2d\sqrt{\frac{2N\mu_2^2}{d} + 1}}\right) \approx \Phi_{\mathcal{N}(0,1)}\left(\Psi_{\mathcal{N}(0,1)}^{-1}(\alpha) - N\sqrt{\frac{\mu_2}{2d}}\right). \quad (7)$$

As $d$ increases for fixed $N$, $j_c$ and $j_c^{-1}$ become symmetric and eventually both tend to $\Phi_{\mathcal{N}(0,1)}\left(\Psi_{\mathcal{N}(0,1)}^{-1}(\alpha)\right) = 1 - \alpha$, i.e. the Type-I and Type-II errors are equal and $\mathcal{A}^{\mathrm{SO}}$'s test is no better than random guessing. Intuitively, this can be understood by considering that the magnitude of $\theta$ (i.e. the test statistic) is influenced less by its individual components as dimensionality increases [28]. Therefore, at $d > 1$, the GM provides progressively stronger $\lambda$-DR guarantees than for scalar queries. However, as the adversary can leverage their knowledge of the model to design a database which influences only a single coordinate of the weight/gradient vector, we mostly use $d = 1$ as a worst-case scenario below. In practice, such a worst-case database may be unrealistic, and the aforementioned result can thus be applied to obtain tighter bounds on MI success in specific scenarios. We note that, in the DPTM, the trade-off function of the GM is always independent of $d$ and has the form $f_{\mathrm{GM}}^{\otimes N} = \Phi_{\mathcal{N}(0,1)}\left(\Psi_{\mathcal{N}(0,1)}^{-1}(\alpha) - \mu\right)$,[3] $\mu = \sqrt{N}\Delta_2/\sigma$, $\otimes N$ denotes $N$-fold composition. The similarity between $f_{\mathrm{GM}}$ and Equation (7) is a consequence of the central limit theorem-like phenomenon [7]. Despite the similarity between the functional forms, we stress that this does not mean that the asymptotic $\lambda$-DR guarantee implies a GDP guarantee, although the converse holds.

---

[3]We say that this mechanism satisfies $\mu$-Gaussian DP (GDP).

**Poisson-Subsampled Gaussian Mechanism (SGM)**   The SGM is relevant in private deep learning [15]. On a $d$-dimensional function $q$ with global $\ell_2$-sensitivity $\Delta_2$, the SGM with diagonal covariance matrix $\sigma^2 \mathbf{I}^d$ samples a mask $\boldsymbol{S} \sim \text{Ber}(p)^d$, where $\boldsymbol{S} \in \{0,1\}^d$ and $\text{Ber}(p)$ denotes a Bernoulli distribution with probability $p$ and outputs $q(\boldsymbol{X}) + \boldsymbol{S} \odot \boldsymbol{Z}, \boldsymbol{Z} \sim \mathcal{N}(0, \sigma^2 \mathbf{I}^d)$, where $\odot$ is the Hadamard product. The dominating pairs are $(\mathcal{N}(0, \sigma^2), (1-p)\mathcal{N}(0, \sigma^2) + p\mathcal{N}(\Delta_2, \sigma^2))$ and $((1-p)\mathcal{N}(\Delta_2, \sigma^2) + p\mathcal{N}(0, \sigma^2), \mathcal{N}(\Delta_2, \sigma^2))$ [22]. Although we have shown in Lemma 3 that the SGM satisfies the MLRP, the test statistic distributions under $\mathcal{H}_0$ and $\mathcal{H}_1$ do not have a tractable form. We thus proceed numerically: First, we instantiate $j_{\text{GM}}$ and $j_{\text{GM}}^{-1}$ for the standard GM as shown above, then use the following general result to amplify the trade-off functions directly.

**Lemma 9.** *Let $T(A, B)(\alpha)$ be a trade-off function between two general distributions $A, B$ representing mechanism outputs. The trade-off functions for the sub-sampled mechanisms are given by $T(A, (1-p)A + pB)(\alpha) = pT(A, B)(\alpha) + (1-p)(1-\alpha)$ and by its inverse.*

We thus obtain the amplified functions $j_p$ and $j_p^{-1}$ from which we compute $J_{\text{SGM}} = \text{C}(j_p, j_p^{-1})$. It follows that the SGM satisfies $J_{\text{SGM}}$-DR. This subsampling technique can also be applied to other mechanisms [22, 23]. An example for the LM is shown in the Appendix. Composition of mechanisms other than the GM is also handled numerically through direct composition of the test statistics as described in [7], Definition 3.1. ff.

## 5   Experimental evaluation

**RTM/DPTM trade-off function comparisons**   Figure 1 shows trade-off functions for the LM, GM and SGM in low privacy (low noise) and high privacy (high noise) settings to compare the error rates in the RTM to the DPTM. The pair of asymmetric DPTM trade-off functions $j, j^{-1}$, is combined into a single symmetric function $J = \text{C}(j, j^{-1})$ (symmetrisation/convexification). We also plot the (symmetrified) trade-off function $f$ for the same mechanism under the DPTM for reference. All RTM curves $(j, j^{-1}, J)$ are closer to the off-diagonal than $f$, indicating that $\mathcal{A}^{\text{SO}}$ has lower MI attack success than $\mathcal{A}^{\text{DP}}$. The "blessing of dimensionality" is shown in subfigure **(f)**: at $d = 30$, stronger $\lambda$-DR is preserved compared to $d = 1$. In the DPTM, the GM enjoys no such privacy amplification.

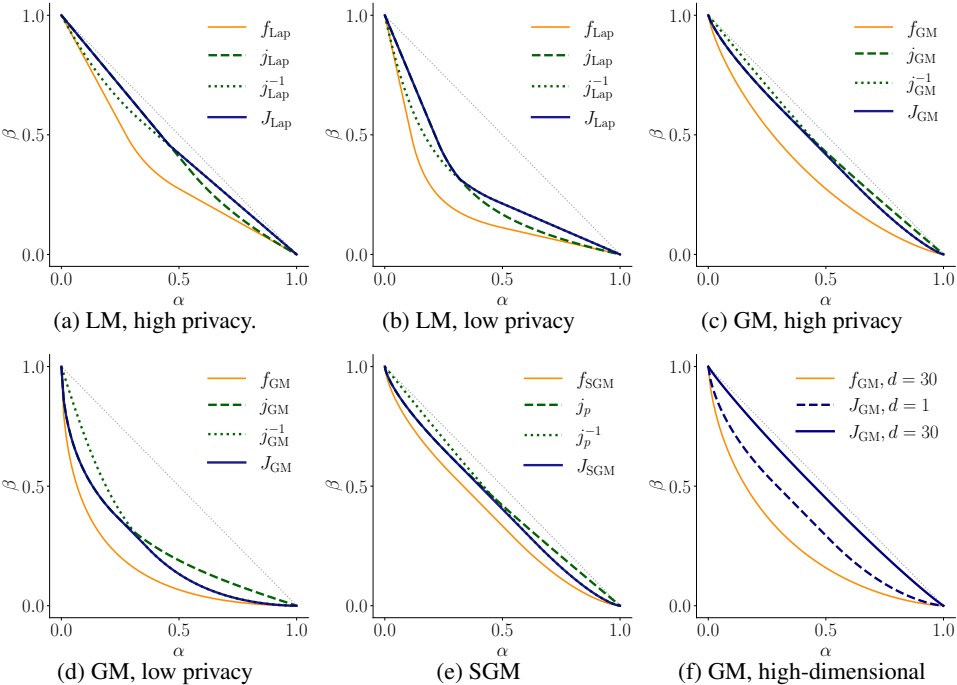

Figure 1: Exemplary trade-off functions for various mechanisms: **(a)** and **(b)**: LM at $\Delta_1/b = 0.6$ and $\Delta_1/b = 1.5$, respectively. **(c)** and **(d)**: GM at $\Delta_2/\sigma = 0.6, d = 1$ and $\Delta_2/\sigma = 1.5, d = 1$, respectively, **(e)**: SGM at $\Delta_2/\sigma = 0.2, d = 1, N = 30, p = 0.4$, **(f)**: GM at $\Delta_2/\sigma = 1$ at $d = 1$ vs. $d = 30$.

**Asymptotic behaviour and power function of the GM**    Figure 2a shows the asymptotic behaviour of a GM with $\triangle_2/\sigma = 1/5, d = 300$ and $N = 150$. The computed trade-off functions become symmetric, approach the off-diagonal and match the plot of the analytic form in Equation (7) exactly, even at a modest number of compositions. Figure 2b plots the power $1 - \beta$ of an MI adversary vs. the $\triangle_2/\sigma$-ratio at a fixed $\alpha = 10^{-3}$ for the GM and Figure 2c for the SGM assuming the worst case of a one-dimensional learning task and a sampling rate of 0.3. The curve for the DPTM in 2c was created using the formula from [29]. As anticipated, the adversary has lower power throughout in the RTM compared to the DPTM, and the loss of power is amplified through subsampling.

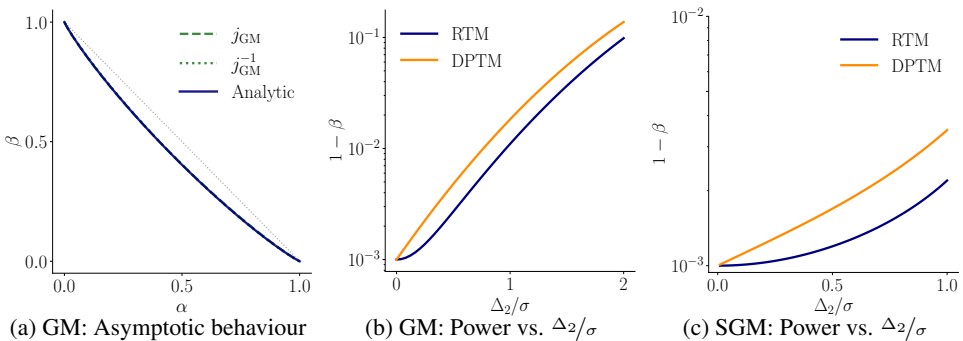

(a) GM: Asymptotic behaviour    (b) GM: Power vs. $\triangle_2/\sigma$    (c) SGM: Power vs. $\triangle_2/\sigma$

Figure 2: Asymptotic behaviour of the GM and power analyses of the GM/SGM.

**MI attacks on DP ML models**    To test the tightness of our theoretical bounds, we now perform an empirical audit using state-of-the-art offline MI attacks on ML models trained with DP. Recall that, despite the nominal difference in the threat model, the offline MI adversary and the RTM adversary compute the same tests and thus have the same capabilities. Each auditing run produces a pair of empirical trade-off functions shown in green and blue. These are compared to the theoretical bound, shown in orange. If the empirical curves crossed below the theoretical one, this would indicate a lack of tightness in our theoretical bounds. All results are shown as averages with standard deviation across 1024 runs for a single step of a one-dimensional synthetic binary classification task, since we are interested in the worst-case scenario under the RTM. Figure 3a shows the results of the LiRa offline MI attack introduced in [12], where $\mathcal{A}^{SO}$ trains shadow models on $D$ and fits a Gaussian likelihood to their confidence scores. At $\triangle_2/\sigma = 1$, this *logit space* auditing technique already nearly perfectly matches the theoretical bound. A further improvement is achieved by implementing the very recent auditing technique presented in [11], also at $\triangle_2/\sigma = 1$. Here, a worst-case gradient with magnitude $\Delta$ ("Dirac canary") is inserted and the hypothesis test takes place in *gradient space* instead of logit space. As seen in Figure 3b, the theoretical curve is matched *exactly* by the empirical one.

So far, we have assumed that $\mathcal{A}^{SO}$ can leverage their knowledge of the model to design a worst-case database which influences only a single coordinate of the gradient. This is useful for auditing purposes, but does not necessarily mirror real-life training. We therefore also repeat the gradient space attack against a linear neural network with a 50-dimensional latent space trained on the *diabetes* dataset [30]. Figure 3c shows that, when the adversary cannot reduce the effective dimensionality of the gradient, the hypothesis test loses substantial power compared to $d = 1$ (dashed line) at $\triangle_2/\sigma = 2$ and the trade-off functions become symmetric, exactly verifying Theorem 3.

Additional results for auditing the LM and SGM are shown in the Appendix.

**Benefits for ML model training**    We next demonstrate that the tighter privacy bounds of the RTM translate to tangible improvements in terms of accuracy/privacy trade-offs in deep learning applications. We performed experiments on three classification tasks: CIFAR-10 using ResNet-9, ImageNet using ResNet-18, and Stanford SNLI using a BERT transformer. In each case, we fixed a final privacy guarantee (in terms of $(\varepsilon, \delta(\varepsilon))$ or equivalently $(\alpha, \beta(\alpha))$) for the DPTM and then trained a set of models up to this privacy guarantee. We then computed the DP-SGD noise scale corresponding to the same hypothesis testing privacy guarantee for the RTM, and re-trained the models for the same number of steps. For all datasets, the enhanced privacy guarantees of the RTM allowed us to obtain models with higher out-of-sample accuracy (e.g. up to $14\%$ higher on CIFAR-10 at $\varepsilon \approx 1$) for the nominally same membership inference guarantee by lowering the DP-SGD noise scale. These results are illustrated in Figure 4 and experimental details can be found in the Appendix.

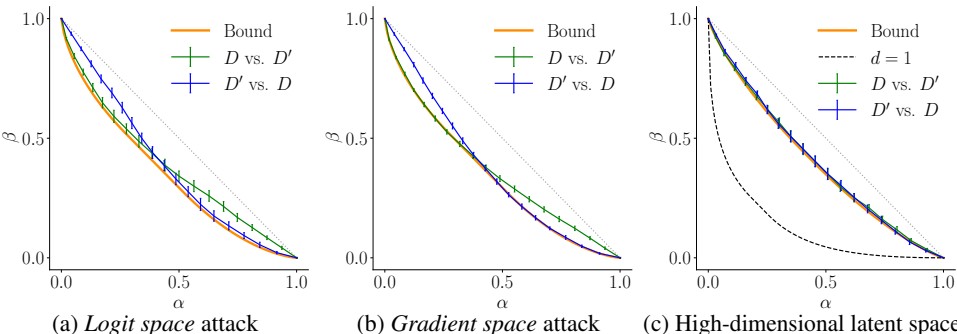

(a) *Logit space* attack     (b) *Gradient space* attack     (c) High-dimensional latent space

Figure 3: Auditing our theoretical bounds using offline MI attacks.

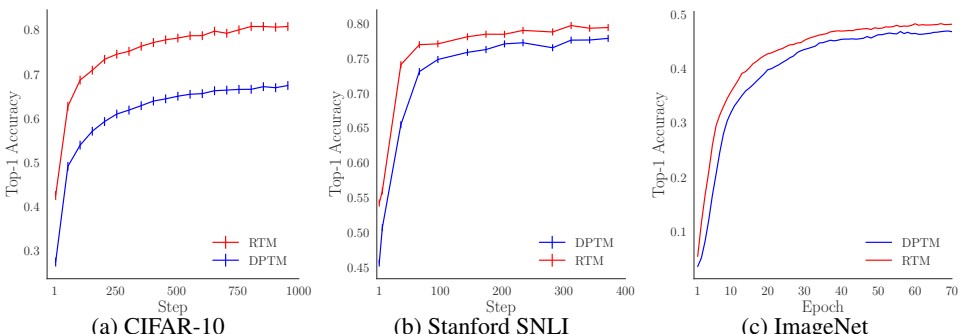

(a) CIFAR-10     (b) Stanford SNLI     (c) ImageNet

Figure 4: Deep learning with DP-SGD. Calibrating the noise scale to the RTM privacy guarantee results in substantially improved Top-1 validation set accuracy compared to calibrating to the DPTM guarantee. (a) CIFAR-10: $0.81 \pm 0.01$ vs. $0.67 \pm 0.01$ at $(1, 10^{-5})$-DP, (b) SNLI: $0.80 \pm 0.005$ vs. $0.78 \pm 0.005$ at $(1, 1.8 \cdot 10^{-6})$-DP, (c) ImageNet: $0.48$ vs. $0.46$ at $(10, 10^{-6})$-DP. Error bars denote standard deviation across $5$ runs.

**Improved bounds on data reconstruction attacks**    As discussed above, successfully defending against MI attacks implies that all weaker attacks such as data reconstruction will also be unsuccessful. In the Appendix, we demonstrate that, in the RTM, data reconstruction adversaries also suffer a diminished success rate compared to the DPTM, as measured in terms of Reconstruction Robustness [31, 32], which can be bounded directly through its relationship to hypothesis testing [33].

## 6   Conclusion

In this work, we consider a threat model relaxation of high practical relevance, where the adversary does not have access to the candidate record in question when performing an MI attack. Our results provide a complete characterisation of the adversary's optimal error rates under the restrictions of the RTM, including at a fixed low Type-I error rate which is particularly meaningful for privacy-sensitive applications such as medical or financial data [12]. Moreover, they allow for a fair and direct comparison between the privacy properties of noise mechanisms under different threat models. Our results can thus guide individuals who train ML systems on sensitive data in the selection of the appropriate noise magnitude to better balance model accuracy vs. privacy protection.

We acknowledge the following limitations. We focused on additive noise mechanisms and the global model of DP in this work. An extension of our results to local DP, mechanisms, mechanisms with discrete outputs and private selection mechanisms is a natural next step. Moreover, our analysis considers the case of non-adaptive composition, which applies to some, but not all, relevant ML tasks. An investigation of adaptive composition scenarios (considering potential correlations between mechanism outputs) would be of interest to expand the scope of our guarantees. Last but not least, we regard the analysis of adversaries who are even more limited in their background knowledge (e.g. no knowledge of the sensitivity, noise magnitude or noise type) as a promising avenue for future work.

## Acknowledgements

This work was supported by a Helmholtz Junior Research Group grant to GK. This work has been funded by the German Federal Ministry of Education and Research and the Bavarian State Ministry for Science and the Arts through the Munich Centre for Machine Learning (MCML) . The authors of this work take full responsibility for its content. This research was supported by the German Ministry of Education and Research (BMBF) under Grant Number 01ZZ2316C (PrivateAIM). This research was supported by the Konrad Zuse School of Excellence in Reliable AI (RelAI).

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

# 7  Supplementary Material

## 7.1  Additional results

**Subsampling and auditing the Laplace Mechanism**  As mentioned in the section on the SGM, numerical subsampling can also be applied to mechanisms other than the GM. Figure 5a shows the trade-off function curves for the LM without subsampling in orange and the subsampled curve for $\Delta_1/b = 1, p = 0.3$ in blue. Similar to the GM, the MI error rates in the RTM (and thus the privacy guarantees) are amplified with sub-sampling.

Figure 5b complements the results on empirical privacy auditing for the LM from the main manuscript. The gradient space auditing technique [11] is again applied to a one-dimensional learning task as described in the main manuscript, however the LM is used instead of the GM. Once more, the empirical privacy audit (green and blue curves) exactly matches the theoretical bounds (shown in orange). As in the main manuscript, curves are averages and standard deviations of 1024 repetitions.

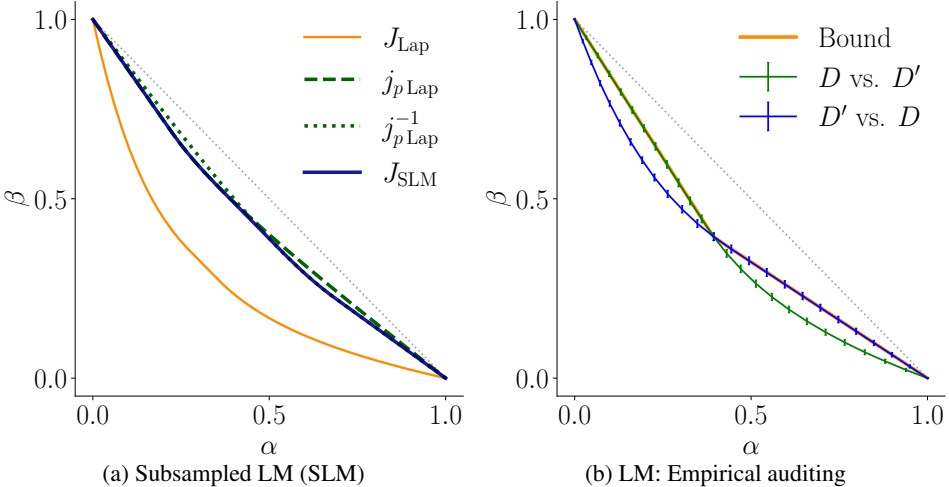

(a) Subsampled LM (SLM)  (b) LM: Empirical auditing

Figure 5: Subsampled Laplace Mechanism and auditing results.

**Auditing the SGM**  Figure 6 shows additional results on privacy auditing when the SGM is used. Here, a single step of DP-SGD with $\Delta_2/\sigma^2 = 1.5$ and $p = 0.3$ is audited using the gradient space auditing technique from [11] as described above. Once more, the empirical privacy audit (green and blue curves) exactly matches the theoretical bounds (shown in orange). As in the main manuscript, curves are averages and standard deviations of 1024 repetitions.

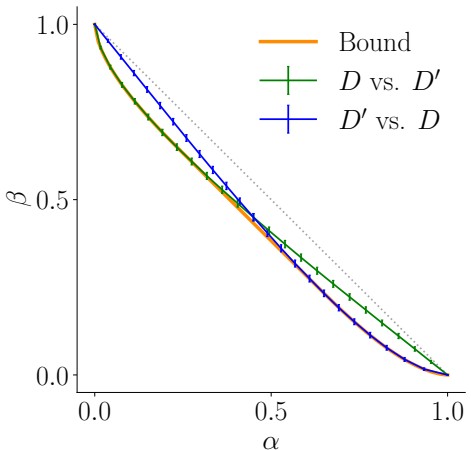

Figure 6: Auditing results for the SGM.

**Bounding data reconstruction** As discussed in the main manuscript, successfully bounding MI implies an ability of the mechanism to also bound all weaker attacks. It is thus possible to translate our MI bounds into a direct and tight bound the probability of a successful data reconstruction attack by using the Reconstruction Robustness (ReRo) framework [31, 32]. In brief, the goal of the ReRo adversary is to obtain a successful reconstruction (from model weights/gradients), defined as a reconstruction loss $\leq \eta$, for example mean squared error or perceptual loss. The ReRo framework models the adversary's prior knowledge as the probability $\kappa(\eta)$ of a successful reconstruction before/without observing the model weights/gradients. Satisfying ReRo then requires the probability of reconstruction after observing the model weights/gradients to be $\leq \gamma$. In summary, a privacy mechanism satisfies $(\eta, \gamma)$-ReRo against an adversary, if, given a prior probability $\kappa(\eta)$ of successful reconstruction (i.e. of reconstruction error $\leq \eta$), the probability of reconstruction after seeing the mechanism's outputs (e.g. model weights or gradients) is no larger than $\gamma$. In recent work, [33] show that, if the trade-off function of a privacy mechanism is known, the probability of reconstruction $\gamma$ can be computed directly, allowing us to extend our theoretical membership inference guarantees to tight bounds on the probability of success of reconstruction attacks. Concretely, it is shown that, if a privacy mechanism's trade-off function $f$ is known, then $\gamma = 1 - f(\kappa(\eta))$. Since $\lambda$-DR is expressed in terms of trade-off functions, the relaxed threat model we study extends naturally to this type of attack. We conducted experiments under (highly) pessimistic assumptions about the adversary: We assume that the adversary has a prior probability of successful reconstruction $\kappa(\eta) = 0.1$ (a more optimistic baseline would be a uniform prior over the entire dataset, for example $\approx 10^{-6}$ for ImageNet). Moreover, following the pessimistic assumption made above, i.e. that the adversary can design a database which influences only a single entry in the weight/gradient vector, we assumed $d = 1$ for the construction of all trade-off functions in the experiments. We then analysed the DP-SGD applications mentioned above (CIFAR-10, ImageNet and Stanford SNLI), whereby we used the Edgeworth expansion technique introduced in [33] to estimate the ReRo upper bound. Even under these permissive assumptions, our findings, detailed in the Figure below, show that the relaxed threat model adversary has a substantially lower probability of success in reconstructing model inputs compared to the DP adversary. For details on the computation of $\varepsilon$, see below.

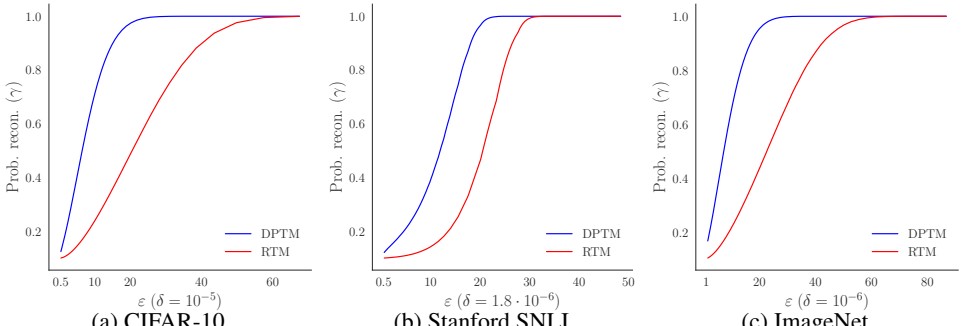

Figure 7: Application to data reconstruction attacks. Panels show the upper bound estimates on the probability of a successful reconstruction attack on the architectures in Figure S1 for increasing privacy budgets $\varepsilon$. Under the assumptions of the RTM, the adversary has a substantially lower probability of reconstruction compared to the DPTM.

## 7.2 Proofs

**Lemma 1.** *In the RTM, no UMP level $\alpha$ test exists for all possible values of $q(D')$, i.e. $\forall \theta \in \Theta \setminus \Theta_0$. Thus, $\mathcal{H}_0$ vs. $\mathcal{H}_1$ (and $\mathcal{H}_1$ vs. $\mathcal{H}_0$) are each not decidable by a single test.*

*Proof.* The claim follows immediately from the Neyman-Pearson Lemma [8]. For a detailed treatment, we refer to Section 3.2. of [25]. □

**Lemma 2.** *The composite LR test $\mathcal{H}_0 : \theta \sim \mathcal{Z}(q(D), \xi)$ vs. $\mathcal{H}_1 : \theta \sim \mathcal{Z}(q(D'), \xi)$ is equivalent to letting $\theta \sim \mathcal{Z}(\mu, \xi)$ and simultaneously conducting two individual one-sided LR tests $r$ and $r'$ with null hypothesis $\mu = 0$ and alternative hypotheses $-\Delta \leq \mu < 0$ for $r$ and $0 < \mu \leq \Delta$ for $r'$, where $\mu = q(D') - q(D)$ and $\Delta$ is the global sensitivity of $q$.*

*Proof.* We will first show that testing $\mathcal{H}_0 : \mathcal{Z}(q(D), \xi)$ vs. $\mathcal{H}_1 : \mathcal{Z}(q(D'), \xi)$ is equivalent to testing $\mathcal{Z}(0, \xi)$ vs. $\mathcal{Z}(q(D') - q(D), \xi)$. We know that $\mathcal{Z}(\mu, \xi)$ is a shift invariant probability measure which admits a density. Thus, for any $c, \theta \in \mathbb{R}$:

$$\mathcal{Z}(\mu + c, \xi)(\theta) = \mathcal{Z}(\mu, \xi)(\theta - c) \Leftrightarrow \mathcal{Z}(\mu - c, \xi)(\theta) = \mathcal{Z}(\mu, \xi)(\theta + c). \tag{8}$$

The LR for $\mathcal{Z}(q(D), \xi)$ vs. $\mathcal{Z}(q(D'), \xi)$ is:

$$\log \frac{\mathcal{Z}(q(D'), \xi)(\theta)}{\mathcal{Z}(q(D), \xi)(\theta)}. \tag{9}$$

Since the LR may be evaluated at an $\theta$, we can evaluate it at $\theta + q(D)$, since $q(D)$ is a known constant:

$$\log \frac{\mathcal{Z}(q(D'), \xi)(\theta + q(D))}{\mathcal{Z}(q(D), \xi)(\theta + q(D))} = \log \frac{\mathcal{Z}(q(D') - q(D), \xi)(\theta)}{\mathcal{Z}(q(D) - q(D), \xi)(\theta)} = \log \frac{\mathcal{Z}(q(D') - q(D), \xi)(\theta)}{\mathcal{Z}(0, \xi)(\theta)}. \tag{10}$$

Therefore, we have shown that testing $\mathcal{Z}(q(D), \xi)$ vs. $\mathcal{Z}(q(D'), \xi)$ is equivalent to testing $\mathcal{Z}(0, \xi)$ vs. $\mathcal{Z}(q(D') - q(D), \xi)$, as the LR for any $\theta$ under both hypothesis testing problems will be the same. Now, let $\mu = q(D') - q(D)$. Since $\mathcal{A}^{\text{SO}}$ does not know the value of $q(D')$, they can only deduct that $\|\mu\| \leq \Delta$. Thus, the two cases $\mu > 0$ and $\mu < 0$ must be considered simultaneously and individually. The claim follows. Observe that the proof is identical for the reversed test $\mathcal{H}_0 : \mathcal{Z}(q(D') - q(D), \xi)$ vs. $\mathcal{H}_1 : \mathcal{Z}(0, \xi)$, i.e. with the null and alternative hypothesis switched. $\qquad \square$

**Lemma 3.** *The individual LR tests $r$ and $r'$ are UMP level $\alpha$ if the noise distribution $\mathcal{Z}$ has the monotone likelihood ratio property (MLRP). The Laplace Mechanism (LM), Gaussian Mechanism (GM) and the Poisson-Subsampled Gaussian Mechanism (SGM) all have the MLPR. Moreover, the power of both tests is maximised when $\|\mu\| = \|q(D') - q(D)\| = \Delta$.*

*Proof.* The existence of an UMP level $\alpha$ LR test for distributions with the MLRP is justified by the Karlin-Rubin Theorem [34], which states that, when the null hypothesis is simple, the alternative hypothesis is one-sided and the LR is a monotone function of the parameter of interest (MLRP), the LR test is UMP level $\alpha$ $\forall \theta \in \Theta \setminus \Theta_0$.

More precisely, the MLRP implies that the LR is non-decreasing or non-increasing in the parameter of interest (in this case $\theta$) of a test. We must consider the cases $\mu < 0$ and $\mu > 0$ individually for each mechanism. Throughout, let $P = \mathcal{Z}(0, \xi)$ and $Q = \mathcal{Z}(\mu, \xi)$. We will replace the placeholder $\mathcal{Z}$ with the distribution of interest as required and consider both tests $P$ vs. $Q$ and $Q$ vs. $P$.

For the Laplace Mechanism (LM), we have:

$$\log \frac{P(\theta)}{Q(\theta)} = \log e^{-\frac{|\theta|}{b}} e^{\frac{|\theta - \mu|}{b}} = \frac{|\theta - \mu| - |\theta|}{b} \quad \text{and} \tag{11}$$

$$\log \frac{Q(\theta)}{P(\theta)} = \log e^{\frac{|\theta|}{b}} e^{-\frac{|\theta - \mu|}{b}} = \frac{-|\theta - \mu| + |\theta|}{b}. \tag{12}$$

For $\mu > 0$, $\log \frac{P(\theta)}{Q(\theta)}$ is non-increasing in $\theta$ and for $\mu < 0$ it is non-decreasing in $\theta$. The opposite holds for $\log \frac{Q(\theta)}{P(\theta)}$ by symmetry.

For the Gaussian Mechanism (GM), we have:

$$\log \frac{P(\theta)}{Q(\theta)} = \log e^{-\frac{\|\theta\|_2^2}{2\sigma^2}} e^{\frac{\|\theta - \mu\|_2^2}{2\sigma^2}} = \frac{\|\theta - \mu\|_2^2 - \|\theta\|_2^2}{2\sigma^2} \quad \text{and} \tag{13}$$

$$\log \frac{Q(\theta)}{P(\theta)} = \log e^{\frac{\|\theta\|_2^2}{2\sigma^2}} e^{-\frac{\|\theta - \mu\|_2^2}{2\sigma^2}} = \frac{-\|\theta - \mu\|_2^2 + \|\theta\|_2^2}{2\sigma^2}. \tag{14}$$

For $\mu > 0$, $\log \frac{P(\theta)}{Q(\theta)}$ is non-increasing in $\theta$ and for $\mu < 0$, it is non-decreasing in $\theta$. Again, the opposite holds for $\log \frac{Q(\theta)}{P(\theta)}$ by symmetry. Observe that the LRs are independent of dimensionality for the (S)GM, hence it suffices to consider the scalar case.

Finally, for the Poisson-Subsampled Gaussian mechanism (SGM), we have:

$$\log \frac{P(\theta)}{Q(\theta)} = \log \left( \frac{1}{pe^{\frac{\theta^2}{2\sigma^2}} - (p-1)e^{\frac{(\theta-\mu)^2}{2\sigma^2}}} \right) + \frac{(\theta-\mu)^2}{2\sigma^2} \quad \text{and} \tag{15}$$

$$\log \frac{Q(\theta)}{P(\theta)} = \log \left( pe^{\frac{\theta^2}{2\sigma^2} - \frac{(\theta-\mu)^2}{2\sigma^2}} - p + 1 \right). \tag{16}$$

The computation here is slightly more involved. Taking the derivatives, we have:

$$\frac{\mathrm{d}}{\mathrm{d}\theta} \log \frac{P(\theta)}{Q(\theta)} = \frac{\mu p e^{\frac{\mu\theta}{\sigma^2}}}{\sigma^2 \left( pe^{\frac{\mu^2}{2\sigma^2}} - pe^{\frac{\mu\theta}{\sigma^2}} - e^{\frac{\mu^2}{2\sigma^2}} \right)} \quad \text{and} \tag{17}$$

$$\frac{\mathrm{d}}{\mathrm{d}\theta} \log \frac{Q(\theta)}{P(\theta)} = \frac{\mu p e^{\frac{\theta^2 - (\theta-\mu)^2}{2\sigma^2}}}{\sigma^2 \left( pe^{\frac{\theta^2 - (\theta-\mu)^2}{2\sigma^2}} - p + 1 \right)}. \tag{18}$$

Both numerators are positive when $\mu$ is positive and negative otherwise. For $\frac{\mathrm{d}}{\mathrm{d}\theta} \log \frac{P(\theta)}{Q(\theta)}$, the denominator is negative when $\mu$ is positive, while for $\frac{\mathrm{d}}{\mathrm{d}\theta} \log \frac{Q(\theta)}{P(\theta)}$, it is negative. Thus, when $\mu$ is positive, $\log \frac{P(\theta)}{Q(\theta)}$ is non-increasing in $\theta$ and non-decreasing otherwise. Conversely, $\log \frac{Q(\theta)}{P(\theta)}$ is non-decrasing with $\mu$ positive and non-increasing otherwise.

So far, we have proven that (1) the LM, GM and SGM have the MLRP and (2) thus, by Karlin-Rubin, there exists an UMP level $\alpha$ test for $\mathcal{Z}(0,\xi)$ vs. $\mathcal{Z}(\mu,\xi)$ as well as for the reverse $\mathcal{Z}(\mu,\xi)$ vs. $\mathcal{Z}(0,\xi)$ for both $r'$ ($\mu > 0$) and $r$ ($\mu < 0$). We note that the proof could have also been achieved by using the fact that the Laplace and Gaussian distributions with known scale are part of a specific one-parameter exponential family, which has the MLRP property. Compare [25], Section 4.2. It remains to show that the power is maximised for $\mu = \Delta$ or $\mu = -\Delta$, i.e. that the power increases with the effect size, in this case $\Delta/\xi$, with $\xi$ assumed fixed.

First, observe that we can reduce our workload by half as, by shift invariance, the tests $r' : \mathcal{Z}(0,\xi)$ vs. $\mathcal{Z}(\mu,\xi)$ and $r : \mathcal{Z}(0,\xi)$ vs. $\mathcal{Z}(-\mu,\xi)$ as well as their reverses are all equivalent. It remains to show that the power function (i.e. $1 - \beta(\mu)$ at a fixed $\alpha$) is maximised at $\mu = \Delta$. In fact, it suffices to show that the power function is strictly increasing as a function of $\mu$, since $\Delta$ is the largest value $\mu$ can take. The power function is a re-parameterised version of the trade-off function for the one-sided UMP level $\alpha$ test (note that this is not the combined test but one of the individual component tests – the combined test is equivalent to performing the two component tests simultaneously and necessarily loses power as it represents multiple testing which increases the false discovery rate). Fix a level $\alpha$.

For the LM, we have [7]:

$$\beta(\mathrm{Lap}(0,b),\mathrm{Lap}(\mu,b) \mid \alpha) = \begin{cases} 1 - e^{\mu/b}\alpha, & \alpha < e^{-\mu/b}/2, \\ e^{-\mu/b}/4\alpha, & e^{-\mu/b}/2 \le \alpha \le 1/2, \\ e^{-\mu/b}(1-\alpha), & \alpha > 1/2. \end{cases} \tag{19}$$

It is easily verified that $1 - \beta(\mathrm{Lap}(0,b),\mathrm{Lap}(\mu,b) \mid \alpha)$ is strictly increasing in $\mu \, \forall \alpha$.

For the GM, we have [7]:

$$\beta\left(\mathcal{N}(0,\sigma^2),\mathcal{N}(\mu,\sigma^2) \mid \alpha\right) = \Phi_{\mathcal{N}(0,1)}\left(\Psi^{-1}_{\mathcal{N}(0,1)}(\alpha) - \mu/\sigma\right), \tag{20}$$

where $\Phi, \Psi^{-1}$ are the CDF and and iSF, respectively. Recall that the function is defined on $[0,1] \times [0,1]$. Taking the derivative with respect to $\mu$ and observing that $\Psi^{-1}(x) = \Phi^{-1}(1-x)$ we obtain:

$$\frac{\mathrm{d}}{\mathrm{d}\mu}\Phi\left(\Phi^{-1}(1-\alpha) - \mu/\sigma\right) = g\left(\Phi^{-1}(1-\alpha) - \mu/\sigma\right)\frac{\mathrm{d}}{\mathrm{d}\mu}\left(\Phi^{-1}(1-\alpha) - \mu/\sigma\right), \tag{21}$$

where $g$ is the PDF of the standard normal. The second term simplifies to:

$$\frac{\mathrm{d}}{\mathrm{d}\mu}\left(\Phi^{-1}(1-\alpha) - \mu/\sigma\right) = -\frac{1}{\sigma}. \tag{22}$$

Thus, we obtain:

$$\frac{\mathrm{d}}{\mathrm{d}\mu} \Phi\left(\Phi^{-1}(1-\alpha) - \mu/\sigma\right) = -\frac{1}{\sigma}g\left(\Phi^{-1}(1-\alpha) - \mu/\sigma\right). \tag{23}$$

Since $\sigma$ is a positive constant and $g(x) > 0 \; \forall x$, the derivative is negative for all values of $\mu$. Thus, $\beta\left(\mathcal{N}(0,\sigma^2), \mathcal{N}(\mu,\sigma^2) \mid \alpha\right)$ is decreasing in $\mu$ and thus $1 - \beta$ is increasing in $\mu$ as required.

Finally, for the SGM, we can invoke Lemma 9 below and re-use the proof for the GM. Fix $0 < p < 1$. Lemma 9, specialised to the SGM, states:

$$\beta\left(\mathcal{N}(0,\sigma^2), p\mathcal{N}(\mu,\sigma^2) + (1-p)\mathcal{N}(0,\sigma^2) \mid \alpha\right) = p\Phi_{\mathcal{N}(0,1)}\left(\Psi_{\mathcal{N}(0,1)}^{-1}(\alpha) - \mu/\sigma\right) + (1-p)(1-\alpha). \tag{24}$$

The second term is independent of $\mu$, thus the claim follows from the proof for the GM.

In conclusion, we have shown that, in the RTM, the adversary must conduct two pairs of individual and simultaneous UMP level $\alpha$ LR tests whose power is maximised when $\mu = \pm\Delta$, i.e. at the dominating pair distributions [22]. This conforms to the intuition that the power of the test should increase when the distributions are "further apart", i.e. when the effect size increases. Moreover, identically to the DPTM, it allows us to consider only the dominating pairs when constructing the trade-off functions, since no other test between the distributions of the mechanism on the two databases can have higher power at the same level $\alpha$ than the test(s) between the dominating pair(s). We have thus established that the optimal tests the sub-optimal adversary $\mathcal{A}^{\mathrm{SO}}$ can construct against the aforementioned noise mechanisms in the RTM are UMP level $\alpha$ tests. These tests are necessarily less powerful than the uniformly most powerful tests that $\mathcal{A}^{\mathrm{NP}}$ can construct because they are only optimal for a **subset** of the values of $q(D')$ at a time, whereas the uniformly most powerful tests of $\mathcal{A}^{\mathrm{NP}}$ are optimal for **all** values of $q(D')$ simultaneously. $\qquad\square$

**Lemma 4.** *If the density of $\mathcal{Z}$ is additionally symmetric about the location parameter, the simultaneous individual LR tests $r$ and $r'$ are equivalent to performing a single test using the magnitude of the observation as a test statistic, i.e. letting $\|\theta\| \sim \|\mathcal{Z}(\mu,\xi)\|$ and testing $\mathcal{H}_0 : \|\mu\| = 0$ vs. $\mathcal{H}_1 : \|\mu\| > 0$. The power of this "combined" test is maximised at $\|\mu\| = \Delta$.*

*Proof.* Observe that $\mathcal{H}_1$ can be split up into two hypotheses: $\mathcal{H}_{1a} : \mu > 0$ and $\mathcal{H}_{1b} : \mu < 0$. Now, we can conduct two simultaneous tests $E_1 : \mathcal{H}_0$ vs. $\mathcal{H}_{1a}$ and $E_2 : \mathcal{H}_0$ vs. $\mathcal{H}_{1b}$. The LRs are:

- For $E_1$: $\Lambda_1(\theta) = \log\frac{\mathcal{L}(\theta|\mathcal{H}_{1a})}{\mathcal{L}(\theta|\mathcal{H}_0)}$.

- For $E_2$: $\Lambda_2(\theta) = \log\frac{\mathcal{L}(\theta|\mathcal{H}_{1b})}{\mathcal{L}(\theta|\mathcal{H}_0)}$.

The rejection region for these tests will be $\{\Lambda_1 > c_1\}$ and $\{\Lambda_2 > c_2\}$, where $c_1, c_2$ are the critical values. The distributions for both $\Lambda_1$ and $\Lambda_2$ under $\mathcal{H}_0$ are identical. It thus suffices to consider the numerator of the LR, which, from the MLRP, is monotone in $\theta$. Cancelling the logarithms, we may thus simplify and consider $\Lambda_1 = \theta$ and $\Lambda_2 = -\theta$. To maintain the same level $\alpha$ for all tests, the critical values should be equal: $c = \hat{c}_1 = \hat{c}_2$, where the hat denotes the new critical values after simplification. We remark for completeness that this will render the tests unbiased. Since these tests are UMP and unbiased, they are also called *UMP unbiased* (i.e. the tests with the greatest power in the set of unbiased tests). For further details, we refer to [25], Section 4. For the combined test, the rejection region is thus: $\{\Lambda_1 > \hat{c}_1\} \cup \{\Lambda_2 > \hat{c}_2\} = \{\theta > c\} \cup \{-\theta > c\} = \{|\theta| > c\}$. This proves the first claim. The claim that the power is maximised when $\|\mu\| = \Delta$ follows from the previous observation and the proof to Lemma 3. For the reverse tests, the proof is identical. $\qquad\square$

**Lemma 5.** *For a (not necessarily symmetric) $\mathcal{Z}$ the Generalised Likelihood Ratio Test (GLRT) using the test statistic $\log\frac{\mathcal{L}(\theta|\mathcal{Z}(\hat{\mu},\xi))}{\mathcal{L}(\theta|\mathcal{Z}(0,\xi))}$, where $\hat{\mu}$ is the maximum likelihood estimate of $\mu$ is equivalent to simultaneously conducting $r$ and $r'$. The GLRT's power is maximised at $\hat{\mu} = \Delta$.*

*Proof.* For the concrete problem $\mu = 0$ vs $\|\mu\| > 0$, the GLRT test statistic can be written:

$$\Lambda(\theta) = \log\frac{\sup_{\mu\in\Theta\setminus\Theta_0}\mathcal{L}(\theta|\mathcal{Z}(\mu,\xi))}{\mathcal{L}(\theta|\mathcal{Z}(0,\xi))} \tag{25}$$

For a single observation $\theta$ (i.e. absent other information) the MLE for the mean $\hat{\mu}$ is equal to the observed value of $\theta$. Thus, the GLRT becomes:

$$\Lambda(\theta) = \log \frac{\mathcal{L}(\theta|\mathcal{Z}(\hat{\mu},\xi))}{\mathcal{L}(\theta|\mathcal{Z}(0,\xi))} \tag{26}$$

Compare the GLRT to the two simultaneous tests:

- $r$: The LR test statistic is $\Lambda_1(\theta) = \log \frac{\mathcal{L}(\theta|\mathcal{Z}(\mu,\xi))}{\mathcal{L}(\theta|\mathcal{Z}(0,\xi))}$ for $\mu > 0$;

- $r'$: The LR test statistic is $\Lambda_2(\theta) = \log \frac{\mathcal{L}(\theta|\mathcal{Z}(\mu,\xi))}{\mathcal{L}(\theta|\mathcal{Z}(0,\xi))}$ for $\mu < 0$.

Observe that these are identical to $\Lambda(\theta)$ when the restricted MLE is used, provided the same critical value is chosen. Thus, the tests are equivalent at the same level $\alpha$. The claim that the power is maximised when $\hat{\mu} = \Delta$ then follows from the previous lemma. For the reverse tests, the proof is identical. We note for completeness that the same result could have been achieved using the Wald or Rao tests, which are derived differently, but are statistically equivalent to the GLRT [35]. $\square$

**Lemma 6.** *If a mechanism satisfies $f$-DP, it also satisfies $f'$-DR, with $f'(\alpha) \geq f(\alpha) \ \forall \alpha \in [0,1]$.*

*Proof.* The lemma is essentially a re-stating of Lemma 1 in the language of f-DP. Since a UMP level $\alpha$ test does not exist for all values of $q(D')$, any test that $\mathcal{A}^{SO}$ constructs is less powerful for at least a subset of the parameter space, and thus the resulting trade-off function in the RTM will be on or above (dominating) the trade-off function in the DPTM. An alternative way to look at this is through the post-processing guarantee of f-DP. As an example, consider using the test statistic $\|\theta\|$ as discussed above. As $\theta$ is the output of a mechanism, we can consider it a post-processing step on $\mathcal{M}(D/D')$. For f-DP, it holds that [7]:

$$T(\mathrm{Proc}(\mathcal{M}(D)), \mathrm{Proc}(\mathcal{M}(D'))) \geq T(\mathcal{M}(D), \mathcal{M}(D')) \tag{27}$$

for any trade-off function $T$. The claim follows by setting the LHS to $f'$, which is the trade-off function constructed using the test statistic $\|\theta\|$ in the RTM and the RHS to $f$, which is the trade-off function of the (NP) UMP level $\alpha$ test in the DPTM. $\square$

**Lemma 7.** *If a mechanism satisfies $f$-DP and $f'$-DR with $f' \geq f$, arbitrary post-processing can only deteriorate privacy up to the $f$-DP guarantee.*

*Proof.* This is essentially the contrapositive to Lemma 6. Since f-DP is resilient to post-processing, the worst deterioration of the $\lambda$-DR-guarantee can never exceed the f-DP guarantee. $\square$

Before proceeding with the proofs about the closed-form/numerical expressions of the trade-off functions, we re-state Definition 2 and provide a brief derivation.

**Definition 2** (Trade-off function construction). *Let $P, Q$ be the distributions of a test statistic under $\mathcal{H}_0$ and $\mathcal{H}_1$, respectively and let $\Phi_{P/Q}$ denote their cumulative distribution function (CDF), $\Psi_{P/Q}$ their survival function (SF) and $\Phi_{P/Q}^{-1}, \Psi_{P/Q}^{-1}$ their respective inverses (iCDF, iSF). Then, we define:*

$$T(\alpha) = \Phi_Q(\Psi_P^{-1}(\alpha)) \ \text{ and } \ T^{-1}(\alpha) = \Psi_P(\Phi_Q^{-1}(\alpha)). \tag{1}$$

*Proof.* The proof is standard. We reproduce it here for the purpose of being self-contained. Consider a one-tailed test for a parameter $\theta$. From basic hypothesis testing theory, given a test statistic $\Lambda(\theta)$ and a critical value $c$, $\alpha = \Pr(\Lambda(\theta) > c \mid \mathcal{H}_0)$ and $\beta = \Pr(\Lambda(\theta) < c \mid \mathcal{H}_1)$. Letting $P$ be the distribution of the test statistic under $\mathcal{H}_0$ and $Q$ its distribution under $\mathcal{H}_0$, we have:

$$\beta(c) = \Phi_Q(c) \ \text{ and } \ \alpha(c) = \Psi_P(c). \tag{28}$$

Since the trade-off function is $T : \alpha \mapsto \beta(\alpha)$, we solve the RHS for $c$ and have:

$$c = \Psi^{-1}(\alpha) \Rightarrow T(\alpha) = \Phi_Q(\Psi_P^{-1}(\alpha)). \tag{29}$$

$T^{-1}$ follows by inverting $T$. The same form manifests by considering $T^{-1}$ the trade-off function for the test with the null and alternative hypotheses exchanged (the reverse test). The role of $P$ and $Q$ will be played by the test statistics for the various mechanisms below. We note that, since the functions are inverses of each other, we may arbitrarily designate one as $T$ and the other as $T^{-1}$. $\square$

**Theorem 1.** *Let $\mu_1 = \Delta_1/b$. The LM satisfies $\mathrm{C}(j_{\mathrm{Lap}}(\alpha), j_{\mathrm{Lap}}^{-1}(\alpha))$-DR with:*

$$j_{\mathrm{Lap}}(\alpha) = \begin{cases} -\exp(-\mu_1)\sinh(\log(\alpha)), & \alpha \geq \exp(-\mu_1) \\ 1 - \alpha\cosh(\mu_1), & \text{otherwise,} \end{cases} \tag{2}$$

*and*

$$j_{\mathrm{Lap}}^{-1}(\alpha) = \begin{cases} \dfrac{1}{\alpha\exp(\mu_1) + \sqrt{\alpha^2\exp(2\mu_1)+1}}, & \alpha < 1/2 - 1/2\exp(-\mu_1) \\ -\dfrac{\alpha-1}{\cosh(\mu_1)}, & \text{otherwise.} \end{cases} \tag{3}$$

*Proof.* $j(\alpha)$ is the trade-off function corresponding to the test $r : \mathrm{Lap}(0,b)$ vs. $\mathrm{Lap}(\Delta_1, b)$ and $j^{-1}(\alpha)$ is the trade-off function for $r' : \mathrm{Lap}(\Delta_1, b)$ vs. $\mathrm{Lap}(0, b)$. Since the density of the Laplace distribution is symmetric, we can invoke Lemma 4 and use the absolute value of the observation as a test statistic.

For $r$, under $\mathcal{H}_0$, the test statistic $\|\theta\|$ follows an exponential distribution with scale parameter $b$. Under $\mathcal{H}_1$, the test statistic $\|\theta\|$ follows a folded Laplace (FoldL) distribution [27] with mean $\Delta_1$ and scale parameter $b$. For $r'$, the hypotheses and distributions are reversed. To construct the trade-off functions, we require the CDF $\Phi$, and iCDF $\Phi^{-1}$. From these, we can construct $\Psi(x) = 1 - \Phi(x)$ and $\Psi^{-1}(x) = \Phi^{-1}(1-x)$. All are available for both distributions in closed form. We have:

$$\Phi_{\mathrm{Exp}(b)}(x) = 1 - \exp(-x/b),$$
$$\Phi_{\mathrm{Exp}(b)}^{-1}(x) = -b\log(1-x)$$

and

$$\Phi_{\mathrm{FoldL}(\Delta_1,b)}(x) = \begin{cases} \exp(-\Delta_1/b)\sinh\left(\frac{x}{b}\right), & 0 \leq x < \Delta_1 \\ 1 - \exp(-x/b)\cosh(\Delta_1/b), & x \geq \Delta_1, \end{cases}$$

$$\Phi_{\mathrm{FoldL}(\Delta_1,b)}^{-1}(x) = \begin{cases} b\log\left(x\exp(\Delta_1/b) + \sqrt{x^2\exp(2\Delta_1/b)+1}\right), & 0 \leq x \leq 1/2 - 1/2\exp(-2\Delta_1/b) \\ b\log\left(\cosh(\Delta_1/b)/(1-x)\right), & 1 > x \geq (1/2 - 1/2\exp(-2\Delta_1/b)). \end{cases}$$

$j$ and $j^{-1}$ then follow by substitution in the expressions for the trade-off functions from Definition 2 above:

$$j(\alpha) = \Phi_{\mathrm{FoldL}(\Delta_1,b)}(\Psi_{\mathrm{Exp}(b)}^{-1}(\alpha)) \quad \text{and} \quad j^{-1}(\alpha) = \Psi_{\mathrm{Exp}(b)}(\Phi_{\mathrm{FoldL}(\Delta_1,b)}^{-1}(\alpha)). \tag{30}$$

$\square$

**Theorem 2.** *Let $\mu_2 = \Delta_2/\sigma$. The GM satisfies $\mathrm{C}(j_{\mathrm{GM}}(\alpha), j_{\mathrm{GM}}^{-1}(\alpha))$-DR with:*

$$j_{\mathrm{GM}}(\alpha) = \Phi_{\chi_d^2(\mu_2^2,\sigma^2)}\left(\Psi_{\chi_d^2(0,\sigma^2)}^{-1}(\alpha)\right) \quad \text{and} \quad j_{\mathrm{GM}}^{-1}(\alpha) = \Psi_{\chi_d^2(0,\sigma^2)}\left(\Phi_{\chi_d^2(\mu_2^2,\sigma^2)}^{-1}(\alpha)\right). \tag{4}$$

*Proof.* The proof is similar to Theorem 1. $j(\alpha)$ is the trade-off function corresponding to the test $r : \mathcal{N}(0,\sigma^2)$. vs. $\mathcal{N}(\Delta_2, \sigma^2)$ and $j^{-1}(\alpha)$ corresponds to the test $r' : \mathcal{N}(\Delta_2,\sigma^2)$ vs. $r : \mathcal{N}(0,\sigma^2)$. The Gaussian density is also symmetric, hence we can appeal to Lemma 4 and use the magnitude of $\theta$ as the test statistic once more. In fact, we will use the squared magnitude for numerical reasons, which does not change the monotonicity behaviour of the test statistics.

For $r$, under $\mathcal{H}_0$, the test statistic $\|\theta\|^2$ follows a central chi-squared distribution with $d$ degrees of freedom scaled by $\sigma^2$, i.e. $\chi_d^2(0,\sigma^2)$. Under $\mathcal{H}_1$, the test statistic $\|\theta\|^2$ follows a noncentral chi-squared distribution with $d$ degrees of freedom and noncentrality parameter $\mu_2^2 = \Delta_2^2/\sigma^2$, i.e. $\chi_d^2(\mu_2^2, \sigma^2)$. For $r'$, the hypotheses and distributions are reversed. To construct the trade-off functions, we require the CDF $\Phi$, and iCDF $\Phi^{-1}$. From these, we can construct $\Psi(x) = 1 - \Phi(x)$ and $\Psi^{-1}(x) = \Phi^{-1}(1-x)$ as above. For the central chi-squared distribution, $\Phi$ has an analytic form, but $\Phi^{-1}$ does not. For the non-central chi-squared distribution, $\Phi$ is expressed in terms of the Marcum Q-function [36], while $\Phi^{-1}$ once again has no analytic form. However, high-precision numerical implementations of all the aforementioned functions are widely available. We thus express $j$ and $j^{-1}$ in terms of these abstract functions and implement them numerically throughout. The concrete expressions follow directly from Definition 4. Moreover, for $d = 1$, a closed form is available for $j$, which is shown in the upcoming Corollary. $\square$

**Corollary 1.** *For $d = 1$, $j(\alpha)$ admits the following closed-form representation:*

$$j_{\text{GM}}(\alpha \mid d = 1) = \Phi_{\mathcal{N}(0,1)}\left(\Psi_{\mathcal{N}(0,1)}^{-1}(\alpha/2) - \mu_2\right) - \Psi_{\mathcal{N}(0,1)}\left(\Psi_{\mathcal{N}(0,1)}^{-1}(\alpha/2) + \mu_2\right). \quad (5)$$

*Proof.* For this proof, it helps to picture Definition 2 with the role of the test statistic played once again by $\|\theta\|^2$. Thus, $\alpha = \Pr(\|\theta\|^2 > c^2 \mid \mathcal{H}_0)$ and $\beta = \Pr(\|\theta\|^2 < c^2 \mid \mathcal{H}_1)$. We can take the square of the critical value since the magnitude is always non-negative and we may choose the critical value arbitrarily; it will also simplify the computations below. Recall that $d = 1$ because $\theta$ is now a scalar.

From the proof to Theorem 2, we have: $\alpha(c^2) = \Psi_{\chi_1^2(0,\sigma^2)}(c^2)$ and $\beta(c^2) = \Phi_{\chi_1^2(\mu_2^2,\sigma^2)}(c^2)$. $\Psi_{\chi_1^2(0,\sigma^2)}$ admits an analytical form:

$$\Psi_{\chi_1^2(0,\sigma^2)}(c^2) = 1 - \frac{\gamma\left(\frac{1}{2}, \frac{c^2}{2\sigma^2}\right)}{\Gamma\left(\frac{1}{2}\right)} = 1 - \frac{\sqrt{\pi}\,\text{erf}\left(\sqrt{\frac{c^2}{2\sigma^2}}\right)}{\sqrt{\pi}} = \quad (31)$$

$$= \text{erfc}\left(\frac{c}{\sqrt{2}\sigma}\right), \quad (32)$$

where $\gamma$ is the lower incomplete gamma function, $\Gamma$ the gamma function and $\text{erf}, \text{erfc}$ are the error function and complementary error function of the Gaussian distribution, respectively. We can now exploit the following pattern:

$$\Psi_{\mathcal{N}(0,1)}(k) = \frac{1}{2}\,\text{erfc}\left(\frac{k}{\sqrt{2}}\right), \quad (33)$$

so the term in Equation (31) can be written as $2\Psi_{\mathcal{N}(0,1)}\left(\frac{c}{\sigma}\right)$. Since $\Psi_{\mathcal{N}(0,1)}$ is invertible, we have that $c = \sigma\Psi_{\mathcal{N}(0,1)}^{-1}\left(\frac{\alpha}{2}\right)$.

Similarly, $\Phi_{\chi_1^2(\mu_2^2,\sigma^2)}(c^2)$ admits a closed-form representation. Recall that $\mu_2^2 = \Delta_2^2/\sigma^2$.

$$1 - Q_{M\frac{1}{2}}\left(\sqrt{\frac{\Delta_2^2}{\sigma^2}}, \sqrt{\frac{c^2}{\sigma^2}}\right) = \quad (34)$$

$$= 1 - Q_{M\frac{1}{2}}\left(\frac{\Delta_2}{\sigma}, \frac{c}{\sigma}\right), \quad (35)$$

where $Q_{M\frac{1}{2}}$ is the Marcum Q-function of order $\frac{1}{2}$ [36].

Substituting the expression for $c$ from above, we obtain:

$$1 - Q_{M\frac{1}{2}}\left(\frac{\Delta_2}{\sigma}, \frac{\sigma\Psi_{\mathcal{N}(0,1)}^{-1}\left(\frac{\alpha}{2}\right)}{\sigma}\right) = \quad (36)$$

$$= 1 - Q_{M\frac{1}{2}}\left(\frac{\Delta_2}{\sigma}, \Psi_{\mathcal{N}(0,1)}^{-1}\left(\frac{\alpha}{2}\right)\right). \quad (37)$$

The Marcum Q-function of order $\frac{1}{2}$ also admits a closed form:

$$Q_{M\frac{1}{2}}(a, b) = \frac{1}{2}\left(\text{erfc}\left(\frac{b - a}{\sqrt{2}}\right) + \text{erfc}\left(\frac{b + a}{\sqrt{2}}\right)\right). \quad (38)$$

Using the pattern $\frac{1}{2}\,\text{erfc}\left(\frac{k}{\sqrt{2}}\right) = \Psi_{\mathcal{N}(0,1)}(k)$ as above, we rewrite Equation (38) as:

$$Q_{M\frac{1}{2}}(a, b) = \Psi_{\mathcal{N}(0,1)}(b - a) + \Psi_{\mathcal{N}(0,1)}(a + b). \quad (39)$$

Finally, we substitute the arguments from Equation 37 and obtain:

$$1 - \left(\Psi_{\mathcal{N}(0,1)}\left(\Psi_{\mathcal{N}(0,1)}^{-1}(\alpha/2) - \frac{\Delta_2}{\sigma}\right) + \Psi_{\mathcal{N}(0,1)}\left(\Psi_{\mathcal{N}(0,1)}^{-1}(\alpha/2) + \frac{\Delta_2}{\sigma}\right)\right) = \quad (40)$$

$$= \Phi_{\mathcal{N}(0,1)}\left(\Psi_{\mathcal{N}(0,1)}^{-1}(\alpha/2) - \mu_2\right) - \Psi_{\mathcal{N}(0,1)}\left(\Psi_{\mathcal{N}(0,1)}^{-1}(\alpha/2) + \mu_2\right), \quad (41)$$

which is the desired form and completes the proof. □

**Lemma 8.** *Let* $\mathrm{GM}_a, \mathrm{GM}_b$ *be GMs with noise variances* $\sigma_a^2 \mathbf{I}^d, \sigma_b^2 \mathbf{I}^d$ *on functions with sensitivities* $\Delta_{2a}, \Delta_{2b}$, *respectively. Then, the non-adaptively composed mechanism* GMC *has trade-off functions:*

$$j_{\mathrm{GMC}}(\alpha) = \Phi_{\chi_d^2(\kappa_c, \sigma_c^2)} \left( \Psi_{\chi_d^2(0, \sigma_c^2)}^{-1}(\alpha) \right) \quad \text{and} \quad j_{\mathrm{GMC}}^{-1}(\alpha) = \Psi_{\chi_d^2(0, \sigma_c^2)} \left( \Phi_{\chi_d^2(\kappa_c, \sigma_c^2)}^{-1}(\alpha) \right). \quad (6)$$

*with* $\kappa_c = (\Delta_{2a} + \Delta_{2b})^2 / \sigma_a^2 + \sigma_b^2$ *and* $\sigma_c^2 = \sigma_a^2 + \sigma_b^2 / 4$.

*Proof.* Let $\boldsymbol{\theta}_a, \boldsymbol{\theta}_b \in \mathbb{R}^d$ be uncorrelated mechanism outputs on two independent queries. Since $\mathcal{A}^{\mathrm{SO}}$ passively observes both outputs *before* testing, the test statistic becomes:

$$\Lambda(\boldsymbol{\theta})_c = \|\boldsymbol{\theta}_c\|_2^2 = \left\| \frac{1}{2} (\boldsymbol{\theta}_a + \boldsymbol{\theta}_b) \right\|_2^2 \lesssim c^2. \quad (42)$$

Of note, the reason why the mean of the observations is the correct choice here follows from Lemma 5, as the MLE of the mean of a normal distribution for $> 1$ observation is the sample average. By the additive properties of isotropic Gaussian noise, the distributions of the observations thus are:

$$\mathcal{H}_0 : \boldsymbol{\theta}_c \sim \mathcal{N} \left( \mathbf{0}, \frac{\sigma_a^2 + \sigma_b^2}{4} \mathbf{I}^d \right) \quad \text{vs.} \quad (43)$$

$$\mathcal{H}_1 : \boldsymbol{\theta}_c \sim \mathcal{N} \left( \frac{\Delta_{2a} + \Delta_{2b}}{2}, \frac{\sigma_a^2 + \sigma_b^2}{4} \mathbf{I}^d \right). \quad (44)$$

Under $\mathcal{H}_0$, the test statistic is thus distributed as:

$$\chi_d^2 \left( 0, \frac{\sigma_a^2 + \sigma_b^2}{4} \right) \quad (45)$$

while under $\mathcal{H}_1$ it is distributed as:

$$\chi_d^2 \left( \frac{\left( \frac{\Delta_{2a} + \Delta_{2b}}{2} \right)^2}{\frac{\sigma_a^2 + \sigma_b^2}{4}}, \frac{\sigma_a^2 + \sigma_b^2}{4} \right) = \chi_d^2 \left( \frac{(\Delta_{2a} + \Delta_{2b})^2}{\sigma_a^2 + \sigma_b^2}, \frac{\sigma_a^2 + \sigma_b^2}{4} \right). \quad (46)$$

From here, the proof continues identically to the proof to Theorem 2 above, from which the claim follows. The generalisation to $> 2$ compositions follows inductively. In particular, for $N$ homogeneous compositions (i.e. with identical sensitivity $\Delta_2$ and noise variance $\sigma^2$), the test statistic under $\mathcal{H}_0$ is distributed as:

$$\chi_d^2 \left( 0, \frac{\sigma^2}{N} \right) \quad (47)$$

while under $\mathcal{H}_1$ it is distributed as:

$$\chi_d^2 \left( \frac{N \Delta_2^2}{\sigma^2}, \frac{\sigma^2}{N} \right), \quad (48)$$

which follows directly from the above. We will use this property in the following proof. $\qquad \square$

**Theorem 3** (Blessing of dimensionality in the RTM). *Consider a GM on a function with sensitivity* $\Delta_2$ *and noise variance* $\sigma^2 \mathbf{I}^d$ *such that* $\Delta_2/\sigma \ll 1$. *Let* $\mu_2 = \Delta_2/\sigma$. *As* $d$ *and/or as the number of non-adaptive compositions* $N$ *increase,* $j_{\mathrm{GM}}$ *and* $j_{\mathrm{GM}}^{-1}$ *tend to the common form:*

$$\Phi_{\mathcal{N}(0,1)} \left( \frac{\Psi_{\mathcal{N}(0,1)}^{-1}(\alpha)}{\sqrt{\frac{2N\mu_2^2}{d} + 1}} - \frac{\sqrt{2}N\mu_2^2}{2d\sqrt{\frac{2N\mu_2^2}{d} + 1}} \right) \approx \Phi_{\mathcal{N}(0,1)} \left( \Psi_{\mathcal{N}(0,1)}^{-1}(\alpha) - N\sqrt{\frac{\mu_2}{2d}} \right). \quad (7)$$

*Proof.* To prove the claim, we will appeal to the central limit theorem (CLT). We will use the facts that the mean of the central chi-squared distribution with $d$ degrees of freedom is $d$ and its variance is $2d$. The noncentral chi-squared distribution with $d$ degrees of freedom and noncentrality $\kappa$ has mean $d + \kappa$ and variance $2d + 4\kappa$. Under the CLT, as $N \to \infty$ and/or as $d \to \infty$, we thus have convergence in distribution as follows. Letting $\frac{\sigma^2}{N} = \delta$ (compare Equations (47) and (48)):

$$\chi_d^2(0, \delta) \to \mathcal{N}(\delta d, 2\delta^2 d) \text{ and} \quad (49)$$

$$\chi_d^2(\kappa, \delta) \to \mathcal{N}(\delta(d + \kappa), \delta^2(2d + 4\kappa)). \quad (50)$$

We now proceed as in the proofs above and consider the test statistic under the null and alternative hypotheses and a critical value $c$ We have:

$$\alpha(c) = \Psi_{\mathcal{N}(0,1)}\left(\frac{c - \delta d}{\delta\sqrt{2d}}\right) \Rightarrow c = \Psi_{\mathcal{N}(0,1)}^{-1}(\alpha)\delta\sqrt{2d} + \delta d \tag{51}$$

and

$$\beta(c) = \Phi_{\mathcal{N}(0,1)}\left(\frac{c - \delta(d+\kappa)}{\delta\sqrt{2d+4\kappa}}\right). \tag{52}$$

For the trade-off function, we substitute to obtain:

$$\Phi_{\mathcal{N}(0,1)}\left(\frac{\Psi_{\mathcal{N}(0,1)}^{-1}(\alpha)\delta\sqrt{2d} + \delta d - \delta d - \delta\kappa}{\delta\sqrt{2d+4\kappa}}\right) = \tag{53}$$

$$= \Phi_{\mathcal{N}(0,1)}\left(\frac{\Psi_{\mathcal{N}(0,1)}^{-1}(\alpha)\sqrt{2d} - \kappa}{\delta\sqrt{2d+4\kappa}}\right) = \tag{54}$$

$$= \Phi_{\mathcal{N}(0,1)}\left(\frac{\Psi_{\mathcal{N}(0,1)}^{-1}(\alpha) - \sqrt{\frac{d}{2}}\frac{\kappa}{d}}{\sqrt{1+\frac{2\kappa}{d}}}\right). \tag{55}$$

Substituting $\kappa \leftarrow {}^{N\Delta_2^2}\!/\sigma^2$ (i.e. the degrees of freedom, compare Equation (48)) and separating the terms, we obtain:

$$\Phi_{\mathcal{N}(0,1)}\left(\frac{1}{\sqrt{\frac{2\Delta_2^2 N}{d\sigma^2}+1}}\Psi_{\mathcal{N}(0,1)}^{-1}(\alpha) - \frac{\sqrt{2}\Delta_2^2 N}{2d\sigma^2\sqrt{\frac{2\Delta_2^2 N}{d\sigma^2}+1}}\right). \tag{56}$$

Letting $\mu_2 = {}^{\Delta_2}\!/\sigma$, we obtain the form of the LHS of Equation (7).

To obtain the RHS of Equation (7), we further massage Equation (55). Concretely, we let ${}^{\kappa}\!/d := \zeta$ and Taylor expand the equation around $\zeta = 0$ to obtain:

$$\Phi_{\mathcal{N}(0,1)}\left(\Psi_{\mathcal{N}(0,1)}^{-1}(\alpha) - \zeta\left(\sqrt{\frac{d}{2}} + \Psi_{\mathcal{N}(0,1)}^{-1}(\alpha)\right) + \mathcal{O}(\zeta^2)\right). \tag{57}$$

When $d$ is large, $\mathcal{O}(\zeta^2)$ vanishes and $\sqrt{\frac{d}{2}}$ dominates the term in the parentheses, yielding:

$$\Phi_{\mathcal{N}(0,1)}\left(\Psi_{\mathcal{N}(0,1)}^{-1}(\alpha) - \zeta\sqrt{\frac{d}{2}}\right). \tag{58}$$

Finally, recursively substituting $\zeta \leftarrow {}^{\kappa}\!/d$ and $\kappa \leftarrow {}^{N\Delta_2^2}\!/\sigma^2$, we get:

$$\Phi_{\mathcal{N}(0,1)}\left(\Psi_{\mathcal{N}(0,1)}^{-1}(\alpha) - \frac{N\Delta_2^2}{\sigma^2}\sqrt{\frac{1}{2d}}\right). \tag{59}$$

Letting ${}^{\Delta_2^2}\!/\sigma^2 = \mu_2^2$, we obtain the RHS of Equation (7). The fact that the two trade-off functions become symmetrical follows this equation's functional form, which corresponds to the trade-off function between two Gaussians with the same variance, i.e. $f_{\mathrm{GM}}$. $\qquad\square$

**Lemma 9.** *Let $T(A, B)(\alpha)$ be a trade-off function between two general distributions $A, B$ representing mechanism outputs. The trade-off functions for the sub-sampled mechanisms are given by $T(A, (1-p)A + pB)(\alpha) = pT(A, B)(\alpha) + (1-p)(1-\alpha)$ and by its inverse.*

*Proof.* Of note, this fact was first observed in [7] and also proved in [22]. We provide a more concise proof here for the purpose of self-containedness. Let $\Phi$, $\Psi$ and their inverses be defined for $A$ and $B$ as above. By Definition 2, we have:

$$T(A, B)(\alpha) = \Phi_B(\Psi_A^{-1}(\alpha)), \tag{60}$$

where $T$ is a generic trade-off function, in our case $j$ or $j^{-1}$. We want to show that for a mixture of distributions $A$ and $B$, with mixture coefficient $p \in (0, 1)$:

$$T(A, (1-p)A + pB)(\alpha) = pT(A, B)(\alpha) + (1-p)(1-\alpha). \tag{61}$$

Define the CDF of the mixture distribution $C$ as $\Phi_C = (1-p)\Phi_A + p\Phi_B$. Now, we have:

$$
\begin{aligned}
T(A, C)(\alpha) =& \Phi_C(\Psi_A^{-1}(\alpha)) = \\
=& (1-p)\Phi_A(\Psi_A^{-1}(\alpha)) + p\Phi_B(\Psi_A^{-1}(\alpha)) = \\
=& (1-p)\Phi_A(\Phi_A^{-1}(1-\alpha)) + p\Phi_B(\Psi_A^{-1}(\alpha)) = \\
=& (1-p)(1-\alpha) + pT(A, B).
\end{aligned}
$$

The inverse $T^{-1}$ is not always easy to construct and, in practice, is usually computed numerically. We have thus shown that we can construct the trade-off functions of a sub-sampled mechanism directly from the trade-off functions of the original mechanism, even if none of them has an analytical form. $\qquad\square$

## 7.3 Experimental details

**Privacy auditing experiments**    All experiments were conducted on a single workstation computer equipped with 64 GB of RAM and a 12-core Intel Xeon CPU with a Thermal Design Power of 105 W and implemented in the Python programming language. For the synthetic learning tasks, a dataset of 1025 samples with one feature each, split evenly across two classes, was generated using `scikit-learn`'s `make_classification` method. Of these, 1024 samples were used for shadow model training and a single example ($x^*$) was held out and its label flipped, similarly to the data poisoning technique of [12] to maximise the point's impact on the model's confidence. This process was repeated for 1024 iterations, and shadow models were trained. No fixed random seed was used. Shadow models were trained in `pytorch` using full-batch DP gradient descent for the experiments in the main manuscript and Poisson sampling for the experiment in the Appendix, with a sample-wise $\ell_2$ norm bound of 1, a noise scale of 1 and implemented using a single linear layer with a single unit. Models were trained to convergence on the training database, at which point the training loss was $< 0.1$. In every case, we considered a single step of model training, i.e. disregarded all previous model updates. For the *logit-space* auditing, logits were transformed by applying the logistic sigmoid activation function, converted to confidence scores and a Gaussian likelihood fitted as described in the Likelihood Ratio Attack Section IV.C of [12]. For executing the offline MI attacks, we followed Algorithm 1 of [12] with the modifications described to implement offline MI. In brief, likelihoods were fitted to the so-called *IN* models, i.e. the ones the adversary is assumed to have access to in the RTM. The models' confidence was then measured on the unseen sample $x^*$ to determine its membership status by implementing the one-sided hypothesis test described in [12] for offline MI. To compute the trade-off functions, the test statistic was evaluated at 50 critical values using `scikit-learn`'s ROC functionality, as the ROC and the trade-off function are complements of each other. Results are reported as averages and standard deviations over the $\beta$ at each critical value $\alpha$, whereby the critical values were chosen identically for each repetition. For the *gradient space* auditing, the gradients of the final training iteration, when norms were lowest, were used as a reference "database" ($D$). To these, a gradient with a magnitude of $\Delta_2$, i.e. equal to the clipping threshold was added to simulate the maximal effect of $x^*$, similar to the *Dirac canary* technique described in [11]. Noise was then added to all gradient samples. Then, the magnitude of the gradients was used as a test statistic as discussed in Appendix A.B. of [12], and as proven to correspond to the optimal hypothesis test under the RTM in Lemma 4. Similar to above, the test statistic was evaluated at 50 critical value settings and the ROCs were computed. For the high-dimensional experiment, we utilised the `diabetes` dataset [30], a 10-dimensional regression dataset with 442 samples. The same strategy was followed as in the gradient space auditing section above, but this time, a model with a latent space dimensionality of 50 was implemented. The noise multiplier was set to 0.5. Once again, a single Dirac canary gradient was added to the gradient "database" and the ROCs were computed as above. The experiments did not require a graphics processing unit and required approximately 60 minutes of net computation time on a CPU. At an assumed carbon efficiency of 0.432 kg/kWh, the computation of these experiments caused approximately 0.04 kg of $CO_2$ equivalent emissions.

**Deep learning with DP-SGD**    CIFAR-10 was trained from scratch using the ResNet-9 architecture from [37] and a subset of the techniques presented in [38] (2 augmentations per draw but no expo-

nential moving average on the weights) to $(1, 10^{-5})$-DP. We used a sampling rate of $4\,096/50\,000$, a noise scale of $10.5$ and trained for $1\,000$ steps in total. For SNLI, we fine-tuned the final 3 layers of a 105M-parameter pre-trained BERT Transformer [39] to $(1, 1.8 \cdot 10^{-6})$-DP using a sampling rate of $4\,096/549\,361$, a noise scale of $1.15$ and $405$ steps. For ImageNet, we fine-tuned the final 12 layers of a ResNet-18 architecture pre-trained on Places365 following the exact hyperparameters from [40] to $(10, 10^{-6})$-DP. For DP accounting, we used the Rényi-DP accountant from the Google Differential Privacy library, whereas for acounting for privacy loss in the RTM, we computed the product distributions of the test statistics as described in [7], Definition 3.1. ff. using direct numerical integration to $50$ digits of precision using the mpmath library. To determine the noise scale for training the models under the RTM, we adapted the technique described in [41]. In brief, we determined the upper and lower noise scale values for which the RTM trade-off functions ($\lambda$-functions) optimally "sandwich" the $f$-DP trade-off function and then chose the more pessimistic of the two noise scales to train the model, keeping all other hyperparameters constant. As a point of comparison, if (hypothetically) one would convert the RTM noise scales to the DPTM in terms of $\varepsilon$, this would result in $\varepsilon$-values of $\approx 6, 2.5$ and $19$ at the same $\delta$ for CIFAR-10, SNLI and ImageNet, respectively. We remark that this conversion does not imply that these models satisfy DP for these $\varepsilon$-values, just that a hypothetical $(\varepsilon, \delta)$-analogue of $\lambda$-DR would achieve these $\varepsilon$-values. The true worst-case $(\varepsilon, \delta)$-DP guarantees remain the same in both cases. The experiments were performed on a GPU cluster using NVidia RTX A6000 GPUs and required a total of $212$ GPU hours. At the carbon efficiency assumed above, the computation of these experiments caused approximately $27.5$ kg of $CO_2$ equivalent emissions. All emission computations were computed as described in [42].

### 7.4 Ethics statement

Our study focuses on the relaxation of the differential privacy (DP) threat model and provides formal bounds on membership hypothesis testing error rates. We acknowledge that our work has positive and potential negative ethical implications, and we strive to address these concerns as thoroughly as possible.

Positive implications of our study include the following:

1. Improved privacy guarantees and higher utility in machine learning models, which may encourage privacy sceptics to adopt DP.
2. Acknowledging that the standard DP threat model may not always be applicable in practice and providing a more realistic alternative.
3. Enhancing our understanding of privacy dynamics in systems by introducing relaxations incrementally, which can demonstrate how privacy guarantees improve as the adversary's power decreases.
4. Providing formal bounds, which offer a more rigorous understanding of the sub-optimal adversary's membership inference capabilities in the relaxed threat model.

Despite these positive implications, we recognise potential drawbacks and have identified mitigations for each:

1. Our guarantees may not hold if the assumptions of the relaxed threat model are violated. To mitigate this, the relaxed threat model guarantee should only be reported alongside the full DP guarantee, which holds in the worst case.
2. If stakeholders or users misunderstand the circumstances under which the relaxed threat model's guarantees break down, they may come away with a false sense of security. To address this, we ensure a comprehensive understanding of the threat model through formal characterisation and clear communication of the exact guarantees provided by the mechanism in the relaxed threat model, and the situations in which they don't hold, e.g. post-processing.

