# OpenReview forum: "Optimal privacy guarantees for a relaxed threat model: Addressing sub-optimal adversaries in differentially private machine learning"
_NeurIPS.cc/2023/Conference — NeurIPS 2023 poster_

### Official Review · Reviewer_9uc5 · 2023-06-29

**Soundness:** 3 good
**Presentation:** 3 good
**Contribution:** 3 good
**Rating:** 7
**Confidence:** 4

**Summary:**

The paper proposes a notion of privacy called Detection Resilience, which quantifies success of a membership inference adversary in a relaxed threat model in which the adversary does not have access to the target example. This is in contrast to the standard strong adversary model in Differential Privacy that has this knowledge. The paper provides a series of theoretical results regarding the relaxed threat model, showing how to characterize the optimal adversary under the relaxation, which for common privacy mechanisms boils down to one-sided hypothesis testing. The trade-off curve of the most powerful test can be used to describe privacy similarly to f-privacy. Next, the paper provides expressions that enable to numerically compute the trade-off curves for Laplace, Gaussian, and Subsampled Gaussian mechanisms. Finally, using synthetic setups the paper visualizes the difference between the trade-off curves in the relaxed model and f-privacy, and shows that the trade-off curves of empirical membership inference attacks are close to those predicted by the theory.

**Strengths:**

**A great step towards useful provable privacy guarantees in DP.** The pessimistic nature of privacy guarantees afforded by DP has been a critical barrier to its deployment. The paper proposes a novel notion of privacy in a relaxation of the standard "strong adversary" model. Thus, it should enable to reduce the noise added during private training while keeping reasonable privacy guarantees, i.e. the guarantees within the relaxed model. *This is the first paper that provides such actionable relaxation* that I am aware of. The prior notions of privacy that incorporate background knowledge rely on assumptions about the data distribution, whereas the notion proposed in the paper does not.

**Weaknesses:**

- W1. **The threat model does not correspond to relevant privacy threats.** The proposed relaxation of the adversarial model in DP, although is a great step towards provable guarantees for realistic attack scenarios, does not model an actually practically relevant threat model. This is because it assumes the adversary does not know the challenge example. To the contrary, in many standard models of practical membership inference attacks the adversary starts with a challenge example they want to attack. For example, this is the case in the "offline attack" by Carlini et al. [12] that is cited as the motivation.

  Instead of a realistic setting, I believe the proposed approach models something that is amenable to the analysis using the decision-theoretic tools in a natural extension of the hypothesis testing interpretation of DP (this is not to say this is something bad on its own).

- W2. **No results with DP-SGD.** The paper seems to make most of the steps needed to evaluate privacy-guarantees under composition, e.g., DP-SGD yet does not quite go there. Without these experiments, it is impossible to gauge the utility gains of relaxing the threat model in most machine learning applications.

Nitpicks:
- Clarity:
  - The contributions paragraph could be interpreted as that the paper studies any relaxed threat models.
  - The text does not provide sufficient context for Lemmas 1-3. It would be clearer to exactly define UMP in the context of any given Lemma.
- DR is a rather confusing name
- Line 92, 499, 508: Should be double bars to denote the norm?
- Line 251: Unclear writing
- Lines 103: The signature of the $\mathcal{M}$ function in the definition of the hypotheses is inconsistent with the notation right above.

**Questions:**

* Line 280 "[In adaptive composition] a database and mechanism parameters are fixed ahead of time, and all intermediate results are released to $A_{SO}$, or –equivalently– the components the mechanism’s outcomes are independent of one another." Are these two settings well-known to be equivalent to adaptive composition? Is there a citation?
* If I understand the former setting ("all intermediate results are released to $A_{SO}$"), the composition results are applicable to standard analyses of DP-SGD. Is this right, or are there fundamental barriers to computing the DR guarantees of DP-SGD using the proposed techniques?
* How exactly is the membership inference attack in the experimental section set up? Does it use shadow models? How many? What are the error bars over in Figure 3?

**Limitations:**

- The paper's title reads as an overclaim. The content covers optimal guarantees to **one** particular relaxed threat model, thus "Optimal privacy guarantees against sub-optimal adversaries" is a stretch.
- The paper does not describe what are the particular privacy harms that are protected in the strong adversary model of DP that are not protected by the relaxed notion.

---

> ### Author Rebuttal · Authors · 2023-08-09
>
> We warmly thank the reviewer for the encouraging and insightful review of our paper. We are truly heartened by the recognition of our work as "a great step towards useful provable privacy guarantees in DP".
>
> > W1. The threat model does not correspond to relevant privacy threats.
>
> We understand the reviewer’s concern that our proposed threat model does not correspond to an actual privacy threat, and would like to clarify this point.
>
> The guarantees we derive are directly applicable to offline MI because in both our membership inference game and in offline MI, the adversary does not evaluate the likelihood at the challenge example. In our threat model, this is by design, whereas in offline MI it is usually by choice (e.g. to save computation). We chose to formulate our privacy game in this specific manner to align with common practices in formal security games. In such games, adversaries are typically designed to exploit all available means to breach security. It would be unconventional in terms of formal design for an adversary to "choose not to do something". We will clarify this point in the final manuscript and thank the reviewer for bringing this to our attention.
>
> > W2. No results with DP-SGD.
>
> We appreciate this suggestion by the reviewer and have now conducted additional experiments which demonstrate the applicability of our threat model relaxation to the analysis of DP-SGD in both image recognition (CIFAR-10 and ImageNet) and natural language processing with transformers (SNLI). As the reviewer correctly assumed, analysing DP-SGD with the relaxed threat model allows us to obtain models with higher accuracy and/or lower privacy trade-offs (e.g. up to 14% higher accuracy in CIFAR-10). Moreover, in response to suggestions by other reviewers, we have also included an analysis of reconstruction attacks against models trained with DP-SGD, where we show that, in the relaxed threat model, the probability of a successful reconstruction attack is provably lower than in the DP threat model. These results can be found in the attached PDF (Figures S1/S2) and further comments can be found in the “Global Reviewer Comment”.
>
> > Nitpicks - Clarity
>
> We thank the reviewer for calling these to our attention. We will make the requested clarifications in the contribution paragraph, define what a “uniformly most powerful test” is, and clarify the norm notation, the writing in line 251 and the signature of the functions in lines 101 ff.
>
> On the topic of choosing the term “detection resilience”, we chose the term “detection” due to its connection to the notion of "signal detection" in hypothesis testing, which is a fundamental concept underlying our work. We are open to considering alternatives that may better convey the meaning of this concept.
>
> > Questions - adaptive composition in lines 280ff.
>
> We acknowledge that the writing here may have been confusing, and will clarify it in the final manuscript. The settings referred to in lines 280 ff. are indeed consistent with the standard approach used when analysing DP-SGD. Specifically, the mechanism parameters are not chosen adaptively at each round but either kept fixed for all rounds or are set in advance. In other words, any changes made to the privacy parameters are determined ahead of time and not influenced by previous outcomes (this is what was meant by “the components of the mechanism outputs are independent of one another”, i.e. conditional independence of the noise draws). Moreover, it is assumed that no post-processing of the intermediate outcomes takes place. In this context, our result can indeed be applied to DP-SGD, and we provide results on deep learning applications with DP-SGD in the attached PDF (Figure S1).
>
> > Questions - experimental details in Figure 3
>
> We thank the reviewer for their inquiry. The setup for the membership inference attack is described in Appendix 7.3, as it was too extensive to fit in the main manuscript. We utilised auditing techniques as described by Carlini et al. (employing shadow models without data poisoning) and by Nasr et al. (using shadow models with additional gradient poisoning). A total of 1024 shadow models were used in our experiments. The error bars in Figure 3 represent the standard deviation of the Type-II error over 1024 repetitions of the experiment (i.e., 1024 shadow models and 1024 repetitions).
>
> > Limitations - change of title
>
> We thank the reviewer for this feedback and understand the concern that the original title may appear as an overclaim. In light of the reviewer’s suggestion, we would propose the following alternative title: "Optimal Privacy Guarantees for a Relaxed Threat Model: Addressing Sub-Optimal Adversaries in Differentially Private Machine Learning". We of course welcome alternative suggestions by the reviewer.
>
> > Limitations - privacy threats which are protected in the DP threat model vs. the relaxed threat model
>
> We thank the reviewer for this remark. Both differential privacy and detection resilience formally protect against membership inference attacks and, by extension, against all weaker forms of attack, notably data reconstruction attacks (we demonstrate the improvement in reconstruction attack bounds in the new experiments – compare the attached PDF). The difference between the two is that the protection of differential privacy also holds against an adversary who can employ auxiliary information (in form of the challenge example) and/or post-process the mechanism output, whereas detection resilience does not. We will clarify this point in the final manuscript.

---

> > ### Comment · Reviewer_9uc5 · 2023-08-10
> >
> > Thank you for the responses.
> >
> > The explanation of the threat model in this response, and a more detailed one in the response to WnyT, seem to be much more intuitive than what is in the paper. I would recommend to try incorporating it into formalisms or in text.
> >
> > > Optimal Privacy Guarantees for a Relaxed Threat Model: Addressing Sub-Optimal Adversaries in Differentially Private Machine Learning
> >
> > I agree a title such as this one would avoid the overclaim --- as long as "relaxed threat model" is singular it should be good.
> >
> > Following your explanations and the additional experiments with DP-SGD, I will consider increasing the score after the questions to the general response are addressed.

---

> > > ### Author Response · Authors · 2023-08-10
> > >
> > > Thank you for your response. Regarding the threat model, we will make sure to incorporate the clearer explanation into the final version of the paper as suggested, and we are happy to adapt the title as discussed. We have also addressed your general response above.

---

### Official Review · Reviewer_5PdE · 2023-07-05

**Soundness:** 3 good
**Presentation:** 2 fair
**Contribution:** 2 fair
**Rating:** 7
**Confidence:** 3

**Summary:**

In the context of membership inference attacks, the authors consider a relaxation of the differential privacy threat model (RTM) wherein the adversary lacks access to the targeted data record. By adopting the binary hypothesis testing formalism commonly employed in the DP literature, the authors prove that the RTM adversary is weaker than the standard DP adversary. Additionally, they prove that it is possible to design uniformly most powerful tests under the monotone likelihood ratio property. Since the Laplace, Gaussian, and Poisson-subsampled Gaussian mechanisms satisfy this property, the authors are able to find the optimal error rates for these three mechanisms under RTM. Finally, the authors include numerical illustrations using state-of-the-art membership inference attacks.

**Strengths:**

In order to incorporate privacy into real-life applications, it is necessary to calibrate privacy mechanisms against practical threats that relax the differential privacy assumptions. The present paper explores this research direction.

**Weaknesses:**

The problem addressed in this paper is of significant importance; however, the paper falls short in delivering substantial results as many of them appear to be straightforward and intuitive. While some proofs may be laborious, a significant portion of the results can be attributed to classical results in statistics. The introduction of the concept of "lambda-Detection Resilience" adds unnecessary complexity, and the results derived from this notion (Lemmas 6 and 7) lack depth. Furthermore, the provided illustrations are elementary and do not contribute significantly to the overall understanding. Collectively, both the theoretical and empirical contributions lack the desired impact.

**Questions:**

While studying additive mechanisms serves as a valuable initial assessment, the emphasis on these mechanisms alone appears somewhat restricted in scope. To enhance the potential impact of the paper, it would be beneficial for the authors to analyze more complex privacy mechanisms. While conducting such an analysis analytically may present considerable challenges, it may be feasible to develop a principled approach that combines theoretical and empirical methods to address this objective.

**Limitations:**

The authors adequately addressed the limitations of their work.

---

> ### Author Rebuttal · Authors · 2023-08-09
>
> We thank the reviewer for their remarks and appreciate the critical feedback, as it provides an opportunity for us to clarify and strengthen our contributions.
>
> > (...) the paper falls short in delivering substantial results as many of them appear to be straightforward and intuitive (...)
>
> We appreciate the reviewer's recognition of the significance of the problem at hand. Similar to previous works in the field (e.g. Dong et al., Gaussian DP), we leverage results from classical hypothesis testing to formulate our bounds. This is a conscious choice to retain the frame of reference of hypothesis testing DP and render our bounds comparable to $f$-DP. We would argue that the substance of our work lies in specialising the theory of sub-optimal hypothesis to sub-optimal adversaries to obtain closed-form trade-off curves for the class of privacy mechanisms most relevant to ML applications. The impact of our findings is thus both theoretical and practical: Our findings unlock the formal analysis of a class of sub-optimal adversaries which are of substantial interest in prior literature (Carlini et al., Ye et al., etc.) but have not yet been formally investigated. They are also general enough to be expanded upon to allow users to compute membership inference guarantees for other privacy mechanisms and compare them between the DP threat model and the relaxed threat model using the established tool of trade-off functions.
>
> Finally, the application of our paper’s results allows users to obtain more accurate machine learning models for the nominally same privacy guarantee. To strengthen our work’s contributions regarding this point, we have now undertaken a set of additional experiments on deep learning in complex image and language modelling applications, showing that our bounds translate into improved privacy/accuracy trade-offs. Moreover, to further strengthen our work’s impact in terms of the interplay between theoretical and practical methods, we have now also included additional results which demonstrate that our guarantees can be directly translated into stronger bounds on data reconstruction attacks, a point which was also of interest to other reviewers. These results can be found in the attached PDF (Figures S1 and S2) and details can be found in the “Global Reviewer Comment”.
>
> >  The introduction of the concept of "lambda-Detection Resilience" adds unnecessary complexity and (...) (Lemmas 6 and 7) lack depth.
>
> While we acknowledge that the concept of $\lambda$-DR may at first glance seem complex, its introduction is important as it serves as a unified frame of comparison with $f$-DP by paralleling its definition. This is vital to accurately depict membership inference guarantees and to differentiate $\lambda$-DR from $f$-DP—a crucial distinction, as a misinterpretation of our guarantees by users could lead to unintended consequences.
> Lemmas 6 and 7 formally specify the relationship between $f$-DP and $\lambda$-DR mechanisms concerning one of the most critical properties of privacy guarantees: post-processing resilience. They therefore allow readers to understand the relevant properties of our guarantee at a glance.
>
> > (...) the provided illustrations (...) do not contribute significantly to the overall understanding (...)
>
> We appreciate the reviewer’s feedback concerning the illustrations, but would like to offer a slightly different perspective on their role and significance. Our paper is rich in mathematical formulas, and the expressions for the trade-off functions are even more complex than in the DP threat model (e.g. due to the appearance of hyperbolic trigonometric functions in the Laplace mechanism). Therefore, the illustrations are not merely supplementary but convey the actual form of the trade-off function’s graphs. Comparisons in function space are inherently comparisons between infinite-dimensional objects. Without these visual aids, understanding the relationships between trade-off functions in the DP threat model and the relaxed threat model would be very challenging for readers. Furthermore, the usage of trade-off function plots like the ones in our paper is established practice in literature. For example, the work by Dong et al. makes extensive use of trade-off function plots and the works of Carlini et al. and Ye et al. use trade-off function visualisations to compare attack success. We thus believe that the illustrations contribute to the overall understanding of our work while aligning with conventions of the field.
>
> > While studying additive mechanisms serves as a valuable initial assessment the emphasis on these mechanisms alone appears somewhat restricted in scope (...)
>
> We thank the reviewer for the suggestion to extend our analysis to additional privacy mechanisms. While the term "complex mechanisms" may encompass a broad range of concepts, we understand the importance of extending our work to diverse applications. We have already taken steps in this direction by conducting the aforementioned additional experiments in more complex machine learning workflows and in bounding data reconstruction attacks. As noted in the manuscript, the theory we develop is general enough to analyse any privacy mechanism, provided the test statistics for the relaxed threat model are well-defined. Our current analysis is consciously centred on the class of additive noise mechanisms, since they are arguably the most important class of mechanisms for machine learning applications (especially the sampled Gaussian mechanism). This importance is illustrated by seminal work in the field, such as the paper by De et al. The reviewer is correct in observing that the analysis of additional privacy mechanisms is challenging, and we are actively working on overcoming these challenges in our ongoing research. As noted in the conclusion of our paper, we view these areas as fruitful targets for future exploration, aligning with our goal to bridge theoretical insights with real-world implementation.

---

> > ### Comment · Reviewer_5PdE · 2023-08-16
> >
> > Thanks a lot for the clarifications and for conducting experiments in more complex machine learning workflows. Now it is easier to put the contribution of your paper in perspective. I raised my score accordingly.

---

### Official Review · Reviewer_pkta · 2023-07-05

**Soundness:** 3 good
**Presentation:** 3 good
**Contribution:** 3 good
**Rating:** 6
**Confidence:** 2

**Summary:**

The paper studies differentially private machine learning (DPML) against sub-optimal membership inferences attacks (MIAs), i.e., the attack cannot access the exact training data. Similar to DPML against the worst-case MIAs, the paper formally characterizes membership inference capabilities in terms of hypothesis testing errors and validates the hypothesis in synthetic datasets.

**Strengths:**

+ The paper is well motivated and well-written
+ Well-formed theoretical results

**Weaknesses:**

- Only evaluations on synthetic datasets



**Questions:**


I do not have any serious comments on the motivation and theoretical results. My only concern is that the proposed DP mechanism is only evaluated on synthetic datasets and does not test in practical settings. Specifically, what are the results on the real-world dataset and applications that are commonly used in the existing DP literature? Whether it is possible to compare the proposed DP with other DP mechanisms against the worst-case adversary?

Could the proposed DP be generalizable to defend against our privacy attacks such as model inversion/data reconstruction attacks? Any comments or thoughts? Regarding this, I would suggest the authors to clearly state "membership inference adversaries" in the title instead of "adversaries" in general.

**Limitations:**

Yes

---

> ### Author Rebuttal · Authors · 2023-08-09
>
> We thank the reviewer for these thoughtful remarks, and would like to address their concerns as follows.
>
> > My only concern is that the proposed DP mechanism is only evaluated on synthetic datasets and does not test in practical settings.
>
> We are grateful for the reviewer’s suggestion to extend our proposed methods to real-world/large-scale datasets to demonstrate that the tighter privacy bounds of the relaxed threat model allow us to obtain better privacy/accuracy trade-offs. In response to interest from the reviewer and the other reviewers, we have thus now conducted additional experiments, which explore image (CIFAR-10 and ImageNet) and natural language (Stanford SNLI) modelling tasks using DP-SGD. These new results show that it is possible to train deep learning models with higher accuracy when considering the relaxed threat model compared to the DP threat model. For instance, we find that calibrating the DP-SGD noise scale to the guarantees of the relaxed threat model instead of the DP threat model results in up to 14% increased accuracy on CIFAR-10 at $\varepsilon=1$. Further details on these results are provided in the attached PDF document (Figure S1) and the “Global Reviewer Comment”.
>
> The reason behind our choice of relatively low-dimensional and synthetic datasets for the auditing experiments is that they are much easier to attack. This allows us to demonstrate the exact tightness of our bounds in a controlled environment, where the difficulty of the attack is not increased by reasons unrelated to the privacy mechanism (e.g. dimensionality of the dataset, sampling randomness in SGD etc.). Performing auditing on real-life datasets would have a high probability to fail to show the tightness of our bounds due to the natural difficulty of the task. Compare for example the work by De et al. (Table 9), where even auditing with MNIST in the DP threat model leads to very loose membership inference bounds. In this sense, performing the auditing experiments using low-dimensional/synthetic data has a higher probability of revealing a lack of tightness in our bounds, which is desirable in the current study.
>
> > Could the proposed DP be generalizable to defend against our privacy attacks such as model inversion/data reconstruction attacks?
>
> The reviewer’s insight into the potential application of our method to bounding model inversion/data reconstruction attacks is correct. Our approach is indeed applicable to bounding the success of reconstruction attacks thanks to the connection to the concept of “Reconstruction Robustness” which has been established in the recent works by Hayes et al. and Balle et al.. Moreover, Kaissis et al. have recently shown that the hypothesis testing interpretation allows one to directly bound the reconstruction attack success probability of an adversary when the trade-off function is known. Therefore, our membership inference bounds can be translated into bounds on reconstruction attacks. In response to the reviewer’s comment, we have now conducted additional experiments on this aspect. For both image recognition and natural language processing tasks, we show that the probability of a successful reconstruction attack is much lower for a relaxed threat model adversary compared to a DP threat model adversary. We refer to the attached PDF document (Figure S2) for the results, and to the “Global Reviewer Comment” for further details.
>
> > I would suggest the authors to clearly state "membership inference adversaries" in the title instead of "adversaries" in general.
>
> We appreciate the reviewer’s suggestion regarding the title. However, since we now also include evidence that our threat model is applicable to bounding reconstruction attacks, we would propose keeping the title, but are open to feedback from the reviewer.

---

> > ### Comment · Reviewer_pkta · 2023-08-15
> > **Response to authors' rebuttal**
> >
> > Thanks for clarifications and they address most of my comments! I will raise my score. One last comment: can you add more details on why only testing $d=1$ in data reconstruction attacks?

---

> > > ### Author Response · Authors · 2023-08-15
> > > **Clarification on $d=1$**
> > >
> > > Thank you for your response. The reason why we choose $d=1$ is that it represents the worst-case in terms of privacy. Any $d>1$ would result in an even lower probability of successful reconstruction for the adversary. The reason for making this worst-case assumption is that in our threat model (like in DP), the dataset can be created/manipulated by the adversary. Therefore, in theory, the adversary can manipulate the dataset in such a way that only a single entry in the weight or gradient vector changes when the data of one individual is added or removed. This makes the hypothesis test easier for the adversary, because the problem is now effectively one-dimensional (compare also lines 296 ff.). Formally, this is the consequence of Theorem 3 in our paper: As the dimensionality of the hypothesis testing space increases, the hypothesis test becomes more difficult (in other words, detecting a signal in high dimensions is more difficult than in low dimensions). Therefore, it is sensible in terms of privacy analysis to analyse the case of a one-dimensional problem to provide a worst-case guarantee. Relatedly, this is also the strategy followed for example by Nasr et al., Tight Auditing of Differentially Private Machine Learning, 2023 (so-called _Dirac Canary_) or by Nasr, Adversary Instantiation: Lower Bounds for Differentially Private Machine Learning, 2021 (Section IV.G).  We will further emphasise the relevance of $d=1$ in the manuscript.

---

### Official Review · Reviewer_WnyT · 2023-07-26

**Soundness:** 3 good
**Presentation:** 3 good
**Contribution:** 2 fair
**Rating:** 5
**Confidence:** 4

**Summary:**

The paper proposes a relaxation of the DP threat model in which the adversary's goal is to perform membership inference without access to the target example. The goal of this threat model is to reflect the offline setting explored in Carlini et al. 2022, Watson et al. 2021, and Ye et al 2022. It is possible to characterize the performance of the optimal adversary in this threat model, similar to the Neyman Pearson optimality in the full DP threat model. The paper computes these optimal performances for a variety of practical additive noise mechanisms for DP.

**Strengths:**

The paper approaches a well motivated goal - relaxing the DP threat model to understand the power of weaker, more realistic, adversaries.

In their threat model, they are able to derive the optimal attack for a number of mechanisms.

Their techniques allow a model trainer to "fall back" on the provable DPTM guarantees, even if an adversary is more powerful than the RTM guarantee would permit.

The paper is well written, and is careful about a lot of points, e.g. DR is not robust to postprocessing and does not compose. It is also very well cited.

**Weaknesses:**

I find the threat model difficult to motivate. At its core, the threat model is defined in a way where the adversary does not learn any private information from succeeding in the game. All it learns is that some example has been added, but no information about this example aside from membership.

The paper suggests a connection to offline MI attacks, but this doesn't seem right. In offline MI, the adversary still knows the example, they just are unable to fully compute the distribution of M(D'). The paper shows that in their threat model, the adversary can only use the magnitude of the noise, not the direction, as they don't know which direction the modification could have come from. However, in offline MI, the adversary *does* know the direction, because they know the example.

The experimental results are performed on very small datasets or small dimension synthetic data.

**Questions:**

The threat model seems more directly applicable to understanding the performance of reconstruction attacks rather than MI attacks, since reconstruction attacks also match your threat model's assumption that the target example is unknown to the adversary. Do you think your results could be applied to those attacks?

**Limitations:**

These are all well noted in the paper.

---

> ### Author Rebuttal · Authors · 2023-08-09
>
> We thank the reviewer for the thoughtful comments on our work and appreciate the opportunity to clarify the reviewer’s questions.
>
> > I find the threat model difficult to motivate (...)
>
> It appears that there may be a misunderstanding about the implications of the adversary succeeding in the membership inference game. In brief, the membership inference security game involves an adversary attempting to determine if an individual's data was used to train a model, without being able to evaluate the likelihood on a challenge example directly. Therefore, success in the membership game does not mean that the adversary does not learn any private information (see step 4 of the game). On the contrary, the implications of the adversary winning the game are actually identical as in the DP threat model, i.e. a breach of membership privacy. The important difference to the DP threat model is the higher *difficulty* of succeeding in the game. This is because, in the relaxed threat model, no direct evaluation of the example’s likelihood is performed.
>
> > The paper suggests a connection to offline MI attacks, but this doesn't seem right (...)
>
> Indeed, in offline MI, the adversary knows the example but chooses not to compute the distribution (likelihood) of $\mathcal{M}(D')$, usually to save computation. In the relaxed threat model we study, the adversary is unable to compute the distribution of $\mathcal{M}(D')$ due to a lack of access to the point. The important similarity is that, *in both scenarios the distribution of $\mathcal{M}(D')$ is not computed by the adversary*. In terms of a security game, we follow formal convention by defining the adversary as not having access to the point in question instead of stating that the adversary “chooses to not evaluate the distribution at this point”; the latter would be inconsistent with typical formal definitions of adversarial games as they are usually presented in security literature, where the adversary uses all available means to breach privacy.
>
> In summary: (1) the relaxed threat model adversary aligns with the DP adversary in every aspect except the fact that they do not evaluate $\mathcal{M}(D')$; (2) success in the membership inference game has the same implications as in the DP threat model; (2) an offline MI adversary does not evaluate $\mathcal{M}(D')$ by choice, whereas the “formal” adversary does not evaluate $\mathcal{M}(D')$ due to a lack of ability. However, the latter distinction is irrelevant in so far as the effect being identical. Therefore, our membership inference game and offline MI can be theoretically analysed in exactly the same way.
>
> We are grateful for the reviewer's remarks on these point and will clarify the writing in the final version of the paper.
>
> > The experimental results are performed on very small datasets or small dimension synthetic data.
>
> We recognise the reviewer’s interest in large dataset applications of our proposed method to obtain better privacy/accuracy trade-offs. Therefore, and also in response to comments from other reviewers, we have now conducted an additional set of experiments. These showcase the usefulness of our technique for improving privacy/accuracy trade-offs in deep learning for image recognition (CIFAR-10 and ImageNet) and natural language processing (Stanford SNLI). For instance, we show up to a 14% increase in accuracy on CIFAR-10 at $\varepsilon=1$ by calibrating the DP-SGD noise scale to the guarantees of the relaxed threat model vs. the DP threat model. These additional experiments on large datasets provide a more comprehensive view of our method's applicability and advantages. Please refer to the “Global Reviewer Comment” and the attached PDF (Figure S1) with the results.
>
> Regarding the reviewer’s concern about the choice of small datasets in our study, it is important to point out that –for the auditing experiments in the manuscript– using small and low-dimensional datasets is preferable since large or high-dimensional datasets can lead to looseness in the empirical bounds due to the inherent difficulty of the attack in higher dimensions. Compare for example the paper by De et al. (Table 9), where, even in the DP threat model and on the relatively low-dimensional MNIST dataset, the empirical membership bounds are very loose. Therefore, large/high-dimensional datasets could hinder our ability to objectively test the tightness of our guarantees, which would be counterproductive for the purposes of the presented study.
>
> > The threat model seems more directly applicable to understanding the performance of reconstruction attacks (...)
>
> We agree with the reviewer and are grateful for this suggestion. Indeed, reconstruction attacks are a promising target for our analysis and the reviewer’s intuition aligns with recent findings in the field. Specifically, works by Hayes et al., building on previous work by Balle et al., have demonstrated that DP mechanisms also bound reconstruction attacks through the concept of “Reconstruction Robustness”. Moreover, a recent work by Kaissis et al. establishes that the hypothesis testing interpretation can be used to formally and tightly bound the probability of a successful reconstruction attack when the trade-off function is known. Thus, our membership inference bounds can directly be translated into bounds on reconstruction attacks, which are (expectedly), much stronger, as reconstruction is a more difficult task. To address the reviewer’s concerns and suggestion, we have conducted additional experiments on bounding reconstruction attack probabilities for the aforementioned image recognition and natural language processing tasks, and show that the relaxed threat model reconstruction probabilities are much lower than those for the DP threat model. The results can be found in Figure S2 of the attached PDF and more details can be found in the “Global Reviewer Comment”.

---

> > ### Comment · Reviewer_WnyT · 2023-08-14
> > **Clarifying the threat model**
> >
> > Thank you for your rebuttal. I think the new results on reconstruction are interesting, and I'm happy to increase my score to reflect this. However, I still take issue with the framing of the threat model, which has not been clarified by the response:
> >
> > In the first reply, you mention the adversary not being able to evaluate the model on the target example, and in the second reply you mention the adversary not being able to compute M(D'). These are very different things. In the LiRA paper, for example, the offline MI adversary does not compute M(D') but still queries the model on the target example. It is incorrect to frame the RTM as reflecting offline MI.

---

> > > ### Author Response · Authors · 2023-08-15
> > > **Clarifying the threat model**
> > >
> > > Thank you for your response. You are correct in noting that there is a technical difference between the way offline MI is carried out in the Carlini et al. (LiRa) paper and the formal specification of the RTM security game. As you correctly noted, the offline MI adversary uses the challenge example to compute the loss or logits of the model because in the LiRA paper, the authors are not directly attacking the weights or the gradient. This distinction however does not improve the hypothesis testing capabilities of the offline MI adversary. To the contrary, as seen in Figure 3a of our paper, using the logits actually makes the attack a little bit weaker compared to a well-executed attack on the model gradient (Figure 3b), which exactly matches the theoretical bound. It would be possible to change step 2 of the RTM security game to specify that $\theta$ can also be a quantity like the loss or the logits computed either on $D$ or on $D'$, which would correspond to the quantities the adversary uses when doing an offline MI experiment in the LiRa paper. The important similarity between offline MI and the RTM remains in the fact that both adversaries do not make use of the distribution/likelihood of $\mathcal{M}(D')$ to render their hypothesis test more powerful. Both adversaries end up having to decide the membership inference problem based on the hypotheses: $\mathcal{H}_0: \theta \sim \mathcal{M}(D)$ vs. $\mathcal{H}_0: \theta \not \sim \mathcal{M}(D)$ (i.e. not distributed as $\mathcal{M}(D)$), where $\theta$ can be e.g. the model weights, gradients, loss or logits.
> > >
> > > We agree with you that this is a subtle point and that using the term “offline MI adversary” to denote any adversary which does not compute the likelihood on $\mathcal{M}(D')$ is not appropriate. We will make this distinction explicit in the manuscript.

---

### Author Rebuttal · Authors · 2023-08-09

## Global Reviewer Comment

We would like to thank the reviewers for their thoughtful comments and suggestions for further experiments. In response to the feedback, we conducted additional experiments focusing on the two requested aspects: (1) obtaining improved accuracy/utility trade-offs in differentially private stochastic gradient descent (DP-SGD), and (2) applying our bounds to data reconstruction attacks. We outline the findings below and refer to the attached PDF for two additional figures which demonstrate the results.

### Large Datasets and DP-SGD Applications
Some reviewers pointed out the importance of large datasets and showcasing applications to DP-SGD to demonstrate the benefits of our approach. Accordingly, we investigated whether the tighter privacy bounds of the RTM could be utilised to train deep learning models with better accuracy/privacy trade-offs, which we found to be the case.

#### Methodology
We performed experiments on three classification tasks: CIFAR-10 using ResNet-9, ImageNet using ResNet-18, and Stanford SNLI using a BERT transformer. For CIFAR-10, we followed the state-of-the-art approach of De et al. and for ImageNet the approach by Kurakin et al., to obtain results comparable to current literature. In each experiment, we first fixed a final privacy guarantee for the DP threat model, and then trained the model to reach this privacy guarantee. Concretely, we selected an $(\varepsilon, \delta)$-DP guarantee for the final model (equivalently, a maximum allowed probability of a true positive membership attack at a pre-set level of Type-I error). We then computed the DP-SGD noise scale corresponding to the same final privacy guarantee for the relaxed threat model, and re-trained the model for the same number of steps.

#### Findings
For all datasets, the enhanced privacy guarantees of the RTM allowed us to obtain models with higher out-of-sample (validation set) accuracy (e.g. up to 14% higher on CIFAR-10 at $\varepsilon=1$) for the nominally same membership inference guarantee, since we were able to lower the DP-SGD noise scale. These results, illustrated in Figure S1, demonstrate how our theoretical bounds can be leveraged by practitioners to achieve improved privacy/accuracy trade-offs through our proposed threat model relaxation.

### Applicability to Data Reconstruction Attacks
Some reviewers also requested a discussion on whether it is possible to extend our membership inference bounds to data reconstruction attack bounds. In fact, it is possible to directly and tightly bound the probability of a successful data reconstruction attack with the Reconstruction Robustness (ReRo) framework (Balle et al.). In brief, the goal of the ReRo adversary is to obtain a successful reconstruction (from model weights/gradients), defined as a reconstruction loss $\leq \eta$, e.g. MSE or perceptual loss. The ReRo framework models the adversary’s prior knowledge as the probability $\kappa(\eta)$ of a successful reconstruction before/without observing the model weights/gradients. Satisfying ReRo then requires that the probability of reconstruction after observing the model weights/gradients to be $\leq \gamma$. In recent work, Kaissis et al. show that, if the trade-off function of a privacy mechanism is known, the probability of reconstruction $\gamma$ can be computed directly, allowing us to extend our theoretical membership inference guarantees to tight bounds on the probability of success of reconstruction attacks. Since Detection Resilience is expressed in terms of trade-off functions, the relaxed threat model we study extends naturally to this type of attack, as suggested by the reviewers.

#### Findings
We conducted experiments under (highly) pessimistic assumptions about the adversary: We assume that the adversary has a prior probability of successful reconstruction $\kappa(\eta)=0.1$ (a more optimistic baseline would be a uniform prior over the entire dataset, i.e. $1/N \approx 10^{-6}$ for ImageNet). Moreover, following the pessimistic assumption made in lines 296ff. of our manuscript (i.e. that the adversary can design a database which influences only a single entry in the weight/gradient vector), we assumed $d=1$ for the construction of all trade-off functions in the experiments.
We then analysed the DP-SGD applications mentioned above (CIFAR-10, ImageNet and Stanford SNLI). Even under these permissive assumptions, our findings, detailed in Figure S2, show that the relaxed threat model adversary has a substantially lower probability of success in reconstructing model inputs compared to the DP adversary.

---

### References for all rebuttal comments below and the attached PDF

Hayes, J., Mahloujifar, S., & Balle, B. (2023). Bounding Training Data Reconstruction in DP-SGD. arXiv preprint arXiv:2302.07225.

Kurakin, A., Song, S., Chien, S., Geambasu, R., Terzis, A., & Thakurta, A. (2022). Toward training at imagenet scale with differential privacy. arXiv preprint arXiv:2201.12328.

De, S., Berrada, L., Hayes, J., Smith, S. L., & Balle, B. (2022). Unlocking high-accuracy differentially private image classification through scale. arXiv preprint arXiv:2204.13650.

Kaissis, G., Hayes, J., Ziller, A., & Rueckert, D. (2023). Bounding data reconstruction attacks with the hypothesis testing interpretation of differential privacy. arXiv preprint arXiv:2307.03928.

Balle, B., Cherubin, G., & Hayes, J. (2022, May). Reconstructing training data with informed adversaries. In 2022 IEEE Symposium on Security and Privacy (SP) (pp. 1138-1156). IEEE.

Devlin, J., Chang, M. W., Lee, K., & Toutanova, K. (2018). BERT: Pre-training of deep bidirectional transformers for language understanding. In Proceedings of NAACL-HLT, pp. 4171-4186. 2019.

Nasr, M., Hayes, J., Steinke, T., Balle, B., Tramèr, F., Jagielski, M., ... & Terzis, A. (2023). Tight Auditing of Differentially Private Machine Learning. arXiv preprint arXiv:2302.07956.

---

> ### Comment · Reviewer_9uc5 · 2023-08-10
> **Clarifications about new experimental settings**
>
> Could you please clarify the following:
> 1. How exactly did you compute the final DR guarantees of DP-SGD? I.e., numeric composition or CLT?
> 2. What is the cost to calibrating for DR in terms of the increase in classical epsilon?

---

> > ### Author Response · Authors · 2023-08-10
> > **Response to clarification request**
> >
> > Thank you for your questions. Regarding question (1), we used numeric composition to obtain exact DR-guarantees. Regarding question (2), we assume that you mean what the "hypothetical" $\varepsilon$-value would be if we converted the reduced noise scale (i.e. the one that DR was calibrated to) to an $(\varepsilon, \delta)$-DP guarantee at the same value of $\delta$. In this case, the values are: For CIFAR-10 $\varepsilon \approx 6$, for SNLI $\varepsilon \approx 2.5$ and for ImageNet $\varepsilon \approx 19$ at the same $\delta$-values. Please let us know in case we misunderstood the question.

---

> > > ### Comment · Reviewer_9uc5 · 2023-08-15
> > >
> > > I have one additional clarification regarding my question (1): What is the formal result that enables to apply accounting for standard DP to accounting of DR? In other words, how exactly can you apply numeric composition of DP to DR?

---

> > > > ### Author Response · Authors · 2023-08-15
> > > > **Details on numerical composition**
> > > >
> > > > Thank you for your response. The work of Dong et al. (Gaussian Differential Privacy, 2019), Definition 3.1 and its subsequent discussion provides the theoretical basis for composing arbitrary trade-off functions. In brief, their result states that, the trade-off function of a composed mechanism is constructed from the product distributions of the test statistics used to represent the individual mechanisms which are being composed. In the DP threat model, the distributions of these test statistics are the privacy loss distributions, i.e. the test statistics of the Neyman-Pearson optimal hypothesis test evaluated at the dominating pair distributions. In our work, we establish the most powerful hypothesis tests in the relaxed threat model and determine that the dominating pair distributions arise at the same parameter values (global sensitivity and noise scale) as in the DP threat model (see lines 126-157 and 202-213). Moreover, we obtain the representations of the test statistics in Section 4. These are “privacy loss distributions for the relaxed threat model”. Combining these results, we can then reuse the same numerical machinery used to compose mechanisms in the DP threat model to compose mechanisms in the relaxed threat model and obtain the corresponding DR guarantees. In our experiments, we directly computed the product distributions of the composed mechanisms’ test statistics through numerical integration, which is exact up to numerical precision (but very time intensive). As part of future work, we plan to evaluate and adapt the newer, fast numerical accountants (e.g. Gopi et al., Numerical Composition of Differential Privacy, 2021; Doroshenko et al., Connect the Dots: Tighter Discrete Approximations of Privacy Loss Distributions, 2022, etc.) in the relaxed threat model.

---

### Decision · Program_Chairs · 2023-09-21

**Decision:**

Accept (poster)

**Comment:**

This paper proposes to study a weaker adversary than the one considered in standard differential privacy.
This is certainly an important and interesting goal, given that there seems to be a large gap between provable privacy guarantees and actual attacks on privacy.
Many of the reviewers were confused by the paper's threat model and claims, which seem to have been clarified with the rebuttal.
I encourage the authors to incorporate the feedback from the reviewers to better describe their main claims in the final version.